# WHO OWNS THIS SAMPLE: CROSS-CLIENT MEMBERSHIP INFERENCE ATTACK IN FEDERATED GRAPH NEURAL NETWORKS

## ABSTRACT

Graph Neural Networks (GNNs) are increasingly integrated with federated learning (FL) to protect data locality in domains such as social networks, finance, and biology. While membership inference attacks (MIAs) have been widely studied in centralized GNNs, their scope in federated settings remains underexplored. We present `CC-MIA`, a framework that reformulates membership inference in federated GNNs into a cross-client attribution problem, where an adversarial client aims to determine not only whether a node was part of training but also which client owns it. `CC-MIA` operates under a realistic threat model: the adversary is a legitimate participant who observes global updates and can eavesdrop on other clients' gradients, a well-studied vulnerability in recent gradient inversion attacks. To approximate the target data distribution, `CC-MIA` leverages publicly available shadow datasets from the same domain, consistent with established MIA practice. The attack combines shadow-based training for membership inference, gradient inversion to reconstruct client subgraphs, and prototype-based matching to assign nodes to clients. Experiments on six benchmark datasets and five federated schemes show that `CC-MIA` consistently outperforms strong MIA baselines, achieving up to 72.16% improvement in inference accuracy. These results highlight that membership inference in federated GNNs naturally extends to client attribution, underscoring the need for defenses robust to gradient-level and client-level leakage. Codes are available at https://anonymous.4open.science/r/CC-MIA-54C3.

## 1 INTRODUCTION

Graphs are a fundamental data structure used to model pairwise relationships between entities, and they naturally arise in domains such as social networks, citation graphs, molecular structures, recommendation systems, and financial transaction networks. Graph Neural Networks (GNNs) extend deep learning to these non-Euclidean domains and have become the de facto standard for learning from graph-structured data Wu et al. (2020). However, training GNNs often involves sensitive information, such as user profiles, social interactions, or molecular properties, raising critical concerns about privacy leakage through inference attacks Sajadmanesh & Gatica-Perez (2021).

Prior work on GNN privacy has mainly focused on centralized settings, where a single server has access to the entire graph. In this paradigm, adversaries have shown that membership inference Wu et al. (2021), property inference He et al. (2020), and link reconstruction Zhang et al. (2022b) are all feasible, exploiting the relational inductive bias of GNNs to infer private attributes or determine whether nodes and edges were part of the training set.

To mitigate these risks, federated learning (FL) has been adopted to decentralize training and protect data locality across multiple participants Bai et al. (2025). Its integration with GNNs has led to Federated Graph Neural Networks (FedGNNs), which enable collaborative learning without centralizing sensitive subgraphs Liu et al. (2024); Schmierer et al. (2025). Yet, despite its privacy-preserving intent, FL remains vulnerable to adversarial behaviors Rao et al. (2024), including poisoning Zhao et al. (2020b), backdoor Zhang et al. (2021a), membership inference Xie et al. (2021b), and model inversion Wu et al. (2024).

The federated paradigm fundamentally changes the adversary's capabilities compared to centralized training. In FedGNNs, no party observes the full graph; instead, participants exchange local updates or gradients. Recent studies have demonstrated that these seemingly limited observations can still reveal sensitive information Bai et al. (2024); Liu et al. (2025). In particular, gradient transmissions can be intercepted and exploited to reconstruct local training examples, motivating a closer look at how membership inference manifests in federated graph settings.

In this work, we introduce `CC-MIA`, a framework that reformulates membership inference in FedGNNs as a cross-client attribution problem. Unlike conventional MIAs that only determine whether a node is in the training set, our framework enables an adversarial client to infer both membership and client ownership of target nodes. The threat model assumes the adversary is a legitimate participant who observes global updates and may eavesdrop on gradient transmissions, consistent with recent gradient inversion attack literature Zhang et al. (2021b); Anand Sinha et al. (2024). To approximate the target distribution, `CC-MIA` leverages publicly available shadow datasets from the same domain (e.g., citation or social graphs), a standard assumption in MIA research. It then combines three components: shadow-based training for membership inference, gradient inversion to reconstruct subgraphs, and prototype-based matching to attribute nodes to their source clients.

Our main contributions are: ❶ We formalize the problem of cross-client membership inference in FedGNNs, positioning the attacker as a participating client and leveraging shadow datasets to conduct node-level membership inference. ❷ We propose a gradient inversion and prototype-matching strategy that enables client ownership attribution from intercepted updates, revealing how gradients encode structural client-specific information. ❸ We conduct extensive experiments on six benchmark datasets and five federated schemes, demonstrating that `CC-MIA` consistently surpasses strong MIA baselines, achieving up to 72.16% improvement in inference accuracy. These results highlight that membership inference in FedGNNs naturally extends to client attribution and underscore the need for defenses that address both gradient-level and client-level leakage.

## 2 RELATED WORKS

Federated learning (FL), despite its privacy-preserving design, remains vulnerable to membership inference attacks (MIAs), which aim to determine whether specific data samples were used during training Nasr et al. (2019). These attacks pose serious privacy risks—such as revealing medical diagnoses—and can support compliance auditing or serve as precursors to more advanced threats like model extraction Zhu et al. (2019); Melis et al. (2019); Wang et al. (2019). In FL, MIAs have evolved from gradient-based approaches to more advanced methods involving shadow training Zhang et al. (2022a), hyperparameter stealing Li et al. (2022), and feature-based inference Liu et al. (2023); Yan et al. (2022). Recent works have extended MIAs to graph neural networks (GNNs), including node-level attacks using posterior distributions He et al. (2021b), label-only attacks Conti et al. (2022), and subgraph- or graph-level inference Olatunji et al. (2021b); Wu et al. (2021); Zhang et al. (2022b); Liu et al. (2022). Detailed related works are stated in Appendix A.16.

However, FedGNNs introduce new and largely unexplored privacy risks. Most GNN-specific attacks are limited to centralized settings with full graph access, while in FedGNNs, clients hold disjoint and heterogeneous subgraphs, and their relationships are implicitly captured through shared model updates, making traditional attacks less effective. Additionally, no prior work has explored cross-client MIAs, where a malicious client aims to infer not just whether a node was used in training, but also which client it belongs to. We identify this overlooked threat and present the first systematic study of membership and ownership inference in FedGNNs from a client-side adversary perspective.

## 3 PROBLEM DEFINITION

### 3.1 THREAT MODEL

Following existing client-side attacks in federated learning Park et al. (2023); Yang et al. (2023); Sun et al. (2024); Xu et al. (2024), we consider a horizontal federated learning setting where one of the participating clients acts as an attacker. This adversarial client has access to its own local graph data, denoted as $G_a(X_a, A_a)$, where $X_a$ represents the node features and $A_a$ denotes the adjacency of the attacker's subgraph.

The attacker is capable of obtaining the following information during the federated learning process:

❶ **Global Model Updates**: The attacker has access to the global model parameters $\Theta$ after each communication round Bagdasaryan et al. (2020); Bhagoji et al. (2019). ❷ **Subgraph Dataset Category**: Following He et al. (2021a); Olatunji et al. (2021a), the attacker is assumed to be aware of the subgraph dataset's category (*e.g.*, citation, product, social, *etc.*). This enables the attacker to match a shadow dataset $\tilde{G}(\tilde{X}, \tilde{A})$ that closely approximates the data distribution of the target category. ❸ **Gradient Eavesdropping**: Based on Wu et al. (2025); Hu et al. (2021), the attacker possesses eavesdropping capabilities, allowing it to intercept the gradient updates $\mathcal{G} = \{\mathcal{G}_1, \cdots, \mathcal{G}_K\}$ uploaded by other clients to the server. ❹ **Why Upload Gradients?** Under our threat model, uploading gradients to the server for aggregation is safer than directly uploading local updated parameters. If an attacker can eavesdrop on the upload link of clients, exposing model parameters would enable the attacker to infer the posterior probabilities of the local GNN directly Olatunji et al. (2021a). Such exposure significantly increases the risk of MIAs on the training-set.

## 3.2 ATTACK TAXONOMY

Building upon the threat model assumptions, we concentrate on training-set membership inference and client-data identification within federated GNN settings and provide formal definitions.

**Definition 1 (Membership Inference Attack)** *Let $G(X, A)$ be the target graph with nodes $X = \{x_1, \ldots, x_N\}$ and let $\mathcal{F}_\Theta$ denote the global GNN embedding function parametrized by $\Theta$. An attacker possesses a shadow dataset $\tilde{G}(\tilde{X}, \tilde{A})$ drawn from the same category as $G$. By extracting embeddings*

$$\tilde{E} = \mathcal{F}_\Theta(\tilde{X}, \tilde{A}), \tag{1}$$

*the attacker trains a binary classifier $f_w: \mathbb{R}^D \to \{0, 1\}$ with parameters $w$ on $(\tilde{E}, \tilde{m})$, where $\tilde{m}_i = 1$ if $\tilde{x}_i$ is a training member and $\tilde{m}_i = 0$ otherwise. Then for any node $x_i \in X$, the membership inference attack is realized by*

$$f_w\big(\mathcal{F}_\Theta(x_i)\big) \approx \begin{cases} 1, & \text{if } x_i \text{ is used in training}, \\ 0, & \text{otherwise}. \end{cases} \tag{2}$$

**Definition 2 (Client-data Identification)** *Let $\mathcal{G} = \{\mathcal{G}_1, \cdots, \mathcal{G}_K\}$ denote the gradients uploaded by $K$ participating clients during each communication round in a federated GNN framework. Suppose an attacker eavesdrops on $\mathcal{G}$ and employs a reconstruction function $f_{rec}(\cdot)$ to approximate the feature representations $r_k = f_{rec}(\mathcal{G}_k)$ of the subgraphs held by client $k \in \{1, \cdots, K\}$. For a target node $x_i \in X = \{x_1, \cdots, x_N\}$, the attacker infers the ownership by determining the client $k$ that minimizes a similarity measure $\mathcal{D}(\cdot)$. Formally, the ownership inference is given by:*

$$k^* = \arg\min_{k \in \{1, \cdots, K\}} \mathcal{D}(x_i, r_k), \tag{3}$$

*where $\mathcal{D}(\cdot)$ quantifies a similarity function.*

## 4 METHODOLOGY

Based on the motivation discussed in Appendix A.2, we extend the scope of MIAs on GNNs from centralized to federated scenarios. This section details the CC-MIA for FedGNNs, decomposing the attack into two objectives: (i) training-set membership inference and (ii) client-data-ownership identification. The overall architecture is illustrated in Fig 1.

### 4.1 MEMBERSHIP INFERENCE ATTACK

**Local GNN Training.** Given the threat model and Definition 1, the target graph is partitioned by METIS Karypis & Kumar (1998) across $K$ clients as $G(X, A) = G_1(X_1, A_1), \ldots, G_k(X_K, A_K)$, where client $k$ holds subgraph $G_k$ with node features $X_k$ and adjacency $A_k$.

During each federated training round $t$, client $k$ computes its local gradient:

$$\mathcal{G}_k^t = \nabla_W \mathcal{L}\big(\mathcal{F}(X_k, A_k; W^t), Y_k\big), \tag{4}$$

where $\mathcal{F}(\cdot; W)$ is the GNN forward pass and $\mathcal{L}$ the cross-entropy loss over client $k$'s labeled nodes $Y_k$. These gradients $\{\mathcal{G}_k^t\}_{k=1}^K$ are then uploaded to the server.

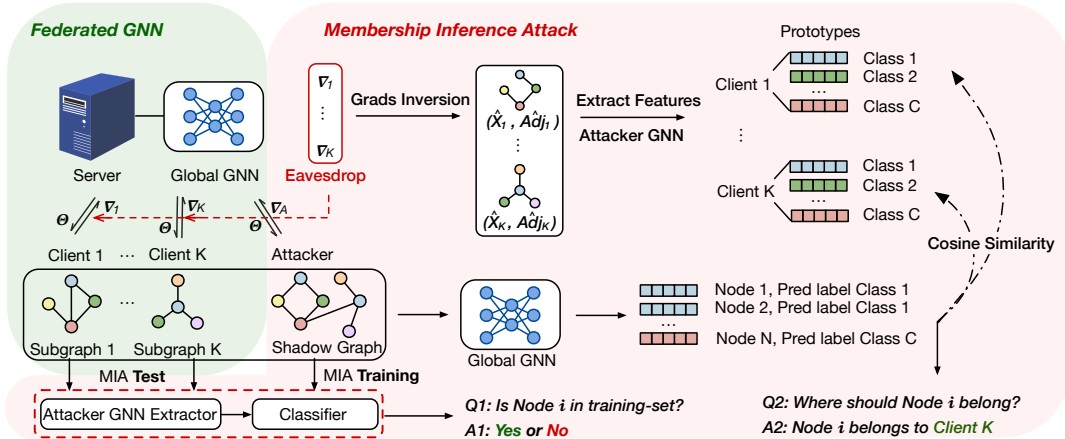

Figure 1: **Framework overview of `CC-MIA`.** Our MIA is designed to address two objectives: (i) Membership inference, where the attacker employs a public shadow dataset with a similar structure to the target data to train a binary classifier, enabling inference of whether a given node belongs to the training set; and (ii) Client-data identification, which employs gradient inversion to reconstruct pseudo node features and adjacencies. The reconstructed features, extracted through the attacker GNN, generate class-specific prototypes for each client. By comparing the node features with these prototypes, CC-MIA determines the client ownership of the nodes. The time and space complexities of `CC-MIA` are discussed in Appendix A.4. The theoretical analysis of the two attacks are shown in Appendix A.5.

**Attacker GNN Training**    The attacker, acting as client $a$, similarly possesses its own local subgraph $G_a(X_a, A_a)$ and additionally acquires a shadow dataset $\tilde{G}(\tilde{X}, \tilde{A})$ drawn from the similar structurally public datasets. Utilizing $\tilde{G}$, the attacker trains an attacker GNN $\tilde{\mathcal{F}}$ to extract features for a downstream classifier. The training process involves iterative gradient updates, formulated as:

$$\tilde{\mathcal{G}}^t \;=\; \nabla_{\tilde{W}}\, \mathcal{L}\big(\tilde{\mathcal{F}}(\tilde{X}, \tilde{A}; \tilde{W}^t),\, \tilde{Y}\big), \quad \tilde{W}^{t+1} = \tilde{W}^t - \eta \Delta \tilde{W}^t \tag{5}$$

**Federated Aggregation**    The server incorporates weight decay and momentum into the gradient aggregation. Denoting the momentum buffer by $M^t$ and a weight-decay coefficient $\lambda$, the update at round $t$ becomes:

$$\Delta W^t = \frac{1}{K} \sum_{k=1}^{K} \mathcal{G}_k^t \;+\; \lambda W^t, M^{t+1} = \mu\, M^t \;+\; \Delta W^t, W^{t+1} = W^t \;-\; \eta\, M^{t+1}, \tag{6}$$

where $\mu \in [0, 1)$ is the momentum coefficient and $\eta$ the global learning rate.

**Feature Extraction.**    $\mathrm{Lap} = \tilde{D}^{-1/2}(\tilde{D} - \tilde{A})\tilde{D}^{-1/2}$ is the normalized laplacian matrix with the degree matrix $\tilde{D}$ and a nonlinear activation $\sigma(\cdot)$. Specially, the attacker gets the 1st-layer attacker GNN embeddings on its shadow graph $(\tilde{X}, \tilde{A})$:

$$\tilde{\mathcal{E}} \;=\; \sigma(\mathrm{Lap}\, \tilde{X}\, W^{(1)} + \mathbf{1}\, b^{(1)\mathsf{T}}), \tag{7}$$

where $W^{(1)} \in \mathbb{R}^{\tilde{D} \times \tilde{D}}$ and $b^{(1)} \in \mathbb{R}^{\tilde{D}}$ are the 1st-layer weights and bias, and $\mathbf{1} \in \mathbb{R}^{|\tilde{X}| \times 1}$ replicates the bias vector. By making the aggregation and embedding operations, we align the formulation with practical FedGNN implementations and clarify the attacker's embedding pipeline.

**Training Classifier.**    A classifier $f_w$ composed of multiple fully connected layers interleaved with batch normalization is employed to conduct membership inference, as follows:

$$H^{(l)} = M^{(l)} \odot \max(B^{(l)}\big(W^{(l)} H^{(l-1)} + b^{(l)}\big), 0) \quad (l = 1, \cdots, L),$$
$$\tilde{\mathcal{Z}} = \mathrm{softmax}\big(W^{(L)} H^{(L-1)} + b^{(L)}\big), \tag{8}$$

we set $H^{(0)} = \tilde{\mathcal{E}}$, where for each layer $l$, $W^{(l)}, b^{(l)}$ are the weight matrix and bias vector. $B^{(l)}(\cdot)$ denotes the affine batch-normalization operator, $M^{(l)} \in \{0, 1\}^{\mathbb{D}}$ is a random binary mask ($\mathbb{D}$ denotes the global training dataset), and $\max(x, 0)$ applies the rectified linear unit elementwise. $\odot$ denotes elementwise multiplication.

The classifier processes feature representations $\tilde{\mathcal{E}}$ that are derived through the shadow GNN, ensuring alignment with the target model's inductive biases and data distribution. This alignment mitigates the risk of overfitting to raw node attributes, which is trained by optimizing the cross-entropy loss over the shadow dataset:

$$\min_{w} \ \mathbb{E}_{i \sim \tilde{G}} \Big[ -\tilde{m}_i \log \sigma\big(f_w(\tilde{\mathcal{E}}_i)\big) - (1 - \tilde{m}_i) \log\big(1 - \sigma\big(f_w(\tilde{\mathcal{E}}_i)\big)\big) \Big], \tag{9}$$

where $f_w(\tilde{\mathcal{E}}_i)$ is the predicted probability of class $c \in \{0, 1\}$. $\tilde{m}_i \in \{0, 1\}$ indicates whether node $i$ belongs to the shadow training-set.

### 4.2 CLIENT-DATA IDENTIFICATION

Given the threat model and Definition 2, eavesdropping on gradients uploaded by clients to the server can infer the ownership of target nodes. This attack contains two stages: (1) gradient inversion, which reconstructs the subgraph associated with each client, and (2) prototype-based client matching, which assigns reconstructed nodes to their respective clients based on specific patterns.

**Eavesdropping**  At each round $t$, eavesdropping is modeled as $\mu_t \sim \text{Bernoulli}(\gamma)$ Wu et al. (2025); Hu et al. (2021), where $\mu_t = 1$ means successful interception. The adversary observes the client update $\xi_t$ if $\mu_t = 1$; otherwise, it estimates $\xi_t$ and the proxy term $\zeta_t$. It knows whether the client was selected ($\delta_t$), and $\mu_t$, $\delta_t$ are independent over time. A local proxy rule is used to simulate updates when interception fails. The estimate $z_t^a$ is updated by interpolating between observed and proxy signals. Estimation accuracy is measured by $\mathbb{E}[\|z_t^c - z_t^a\|^2]$, where $z_t^c$ denotes the true client update.

**Gradient Inversion for Client Subgraph.**  As discussed in the threat model, an attacker can intercept gradients from individual clients through eavesdropping. Specifically, each client's 1st-layer gradients are exploited to reconstruct the node features and the adjacency. Gradients are denoted as $g \in \mathbb{R}^{d_{\text{in}} \times d_{\text{hid}}} = \nabla_W \mathcal{L}$, where $W$ represents the parameter weights of the first GCN layer, and $\mathcal{L}$ denotes the loss of each client GNN.

To reconstruct client data from gradients, it is necessary to generate synthetic gradients, denoted as $\nabla_W \hat{\mathcal{L}}$, and iteratively optimize them to approximate the intercepted real gradients. As discussed in Anand Sinha et al. (2024), this approximation is achieved by minimizing the negative cosine similarity, formulated as:

$$\mathbb{L} = 1 - \frac{\nabla_W \mathcal{L} \cdot \nabla_W \hat{\mathcal{L}}}{\|\nabla_W \mathcal{L}\| \|\nabla_W \hat{\mathcal{L}}\|} \tag{10}$$

The synthetic gradient $\nabla_W \hat{\mathcal{L}}$ is generated by feeding a pseudo-graph input into the GNN. In CC-MIA, we assume that the input labels are known, as they can be easily inferred from the gradients in classification tasks with cross-entropy loss Zhao et al. (2020a). Many real-world graphs, such as social networks, exhibit the property of feature smoothness, where connected nodes tend to have similar attributes. To ensure the reconstructed graph data adheres to this smoothness property, we adopt the smoothness loss proposed in Zhang et al. (2021b) for gradient fitting:

$$\mathbb{L}_{\text{smooth}} = \text{tr}(X^T \hat{\text{Lap}} X) = \frac{1}{2} \sum_{i,j=1}^{N} A_{ij} \left( \frac{x_i}{\sqrt{d_i}} - \frac{x_j}{\sqrt{d_j}} \right)^2, \tag{11}$$

where $\hat{\text{Lap}}$ denotes the normalized laplacian matrix of $A$ and $D$ is the diagonal matrix of $A$, $A_{ij}$ denotes elements of the adjacency, and $d_i$ represents the degree of node $i$.

To enhance the sparsity of the graph structure, the Frobenius norm of the adjacency matrix $A$ is incorporated into the loss function. The final objective function is defined as:

$$\min_{\hat{X},\hat{A}} \quad \hat{\mathbb{L}} = \mathbb{L} + \alpha\mathbb{L}_{\text{smooth}} + \beta\|\hat{A}\|_F^2, \tag{12}$$

where $\|\hat{A}\|_F^2$ denotes the adjacency during optimization. The coefficients $\alpha$ and $\beta$ control the relative importance of the smoothness and sparsity constraints, respectively.

The features of the reconstructed nodes, $\hat{X}$, can be used directly for the downstream task. However, the reconstructed adjacency requires further mapping and sampling. Specifically, the gradient descent step for the adjacency includes a projection step. During this step, each entry of the adjacency $\tilde{A}_{ij}$ is updated using an entry-wise projection operator defined as follows:

$$\hat{A}_{ij} = \text{proj}_{[0,1]}(\tilde{A}_{ij}) = \begin{cases} 1, & \tilde{A}_{ij} > 1 \\ 0, & \tilde{A}_{ij} < 0 \\ \tilde{A}_{ij}, & \text{otherwise} \end{cases}, \tag{13}$$

To refine the adjacency further, we retain only the top $N_e$ edges with the largest weights, setting their values to 1, while all other entries are set to 0:

$$\hat{A}_{ij} = \begin{cases} 1, & \text{if } (i,j) \in E_{\text{top}}, \\ 0, & \text{otherwise}, \end{cases} \tag{14}$$

where $E_{\text{top}}$ denotes the set of $N_e = \rho \cdot N$ edges with the highest weights in $\hat{A}$. $\rho$ represents the known edge density, and $N$ is the number of nodes.

Specifically, `CC-MIA` utilizes Algorithm 1 to reconstruct the client's node features and adjacency.

**Prototype-based Client Matching.** As an adversarial client within the FedGNNs, the attacker gains access to the global GNN $\mathcal{F}$ distributed by the server. By leveraging this model, the attacker feeds a reconstructed graph into $\mathcal{F}$ to generate pseudo-embeddings $\hat{\mathcal{E}}$ for nodes associated with different target clients. These pseudo-embeddings serve as the foundation for constructing class-specific prototypes for each targeted client:

$$\hat{\mathcal{E}}_k = \mathcal{F}_1(\hat{X}, \hat{A}),$$
$$\mathcal{P}_k = \left\{ c \mapsto \frac{1}{|\mathcal{I}_c^k|} \sum_{j \in \mathcal{I}_c^k} \hat{\mathcal{E}}_k^j \,\middle|\, c \in \mathcal{C}_k \right\}, \tag{15}$$

where $\mathcal{C}_k$ is the set of all classes of client $k$. $\mathcal{I}_c^k$ is the sample index set labeled as $c$ in client $k$.

Each prototype set of client $k$ is $\mathcal{P}_k = \left\{ c \mapsto \mu_c^{(k)} \mid c \in \mathcal{C}_k \right\}$, $\mu_c^{(k)} \in \mathbb{R}^D$ represents the prototype vector of class $c$ for client $k$, and $\mathcal{C}_k$ denotes the set of classes for client $k$.

The real node information to be analyzed is passed to the adversary. By measuring node embeddings generated by the GNN 1-st layer for extraction with the class prototypes of each target client:

$$\mathcal{E} = \mathcal{F}_1(X, A),$$
$$d_{i,k} = \begin{cases} 1 - \frac{\mathcal{E}_k^i \cdot \mu_{y_i}^{(k)}}{\|\mathcal{E}_k^i\| \cdot \|\mu_{y_i}^{(k)}\|} & y_i \in \mathcal{C}_k \\ \infty & \text{otherwise} \end{cases}, \tag{16}$$

the target node is assigned to the client with the closest prototype, determined by $\hat{k}_i = \arg\min_{k \in \{1, \cdots, K\}} d_{i,k}$. The complete matching process is detailed in Algorithm 2.

## 5 EXPERIMENTS

### 5.1 BASELINES

In our experiments, we evaluate three representative GNN models, including GCN Kipf & Welling (2017), GAT Velickovic et al. (2018), and GraphSAGE Hamilton et al. (2017) as global models

Table 1: Performance of member inference attacks and client-data identification. Clts: Clients.

| Dataset | Approach | Shadow | HP-MIA | GAN-Based | CS-MIA | CC-MIA | 3-Clts | 4-Clts | 5-Clts | 6-Clts | 7-Clts | 8-Clts | 9-Clts | 10-Clts |
|---|---|---|---|---|---|---|---|---|---|---|---|---|---|---|
| **Client-uniform Probability** | - | - | - | - | - | | 33.33 | 25.00 | 20.00 | 16.66 | 14.29 | 12.50 | 11.11 | 10.00 |
| Cora | FedAvg | DBLP | 51.05 | 53.15 | 57.87 | **82.04** | 59.27 | 55.61 | 56.31 | 38.92 | 34.08 | 33.20 | 27.18 | 27.95 |
| | FedProx | DBLP | 50.79 | 50.42 | 74.72 | **83.31** | 55.80 | 47.56 | 40.66 | 36.15 | 40.47 | 28.66 | 23.67 | 19.13 |
| | SCAFFOLD | DBLP | 52.04 | 52.09 | 69.04 | **83.38** | 64.99 | 42.32 | 34.12 | 35.75 | 41.99 | 35.56 | 25.59 | 22.01 |
| | FedDF | DBLP | 52.29 | 51.89 | 72.29 | **82.71** | 36.74 | 39.73 | 40.07 | 29.87 | 25.70 | 25.48 | 23.49 | 19.46 |
| | FedNova | DBLP | 54.08 | 56.16 | 74.51 | **84.37** | 39.00 | 30.80 | 23.67 | 18.65 | 17.91 | 18.50 | 14.55 | 12.11 |
| Citeseer | FedAvg | PubMed | 51.69 | 50.76 | 72.68 | **86.04** | 51.85 | 48.21 | 44.00 | 36.55 | 35.32 | 32.04 | 31.89 | 22.00 |
| | FedProx | PubMed | 52.57 | 53.56 | 71.10 | **85.98** | 52.15 | 44.66 | 44.18 | 42.44 | 34.39 | 30.36 | 28.13 | 26.45 |
| | SCAFFOLD | PubMed | 49.89 | 51.23 | 59.70 | **85.89** | 55.64 | 45.15 | 42.11 | 40.49 | 39.92 | 33.78 | 29.13 | 23.65 |
| | FedDF | PubMed | 54.32 | 52.67 | 74.61 | **86.83** | 51.61 | 51.19 | 44.39 | 37.60 | 29.37 | 28.70 | 26.39 | 23.35 |
| | FedNova | PubMed | 53.49 | 58.14 | 74.42 | **85.97** | 42.23 | 38.35 | 34.48 | 30.42 | 28.46 | 27.35 | 25.28 | 22.78 |
| PubMed | FedAvg | DBLP | 55.74 | 50.23 | 58.14 | **71.73** | 49.56 | 41.56 | 32.68 | 28.47 | 25.55 | 19.16 | 17.32 | 14.52 |
| | FedProx | DBLP | 54.98 | 54.12 | 58.79 | **71.84** | 46.24 | 32.91 | 26.03 | 23.24 | 22.15 | 21.41 | 15.67 | 12.23 |
| | SCAFFOLD | Physics | 53.75 | 54.78 | 52.68 | **71.25** | 44.12 | 38.83 | 26.10 | 25.15 | 24.34 | 20.64 | 18.94 | 17.85 |
| | FedDF | DBLP | 56.09 | 55.01 | 59.24 | **72.03** | 43.55 | 39.62 | 24.55 | 24.22 | 24.10 | 22.68 | 20.64 | 18.37 |
| | FedNova | DBLP | 55.48 | 52.34 | 57.56 | **65.23** | 45.81 | 40.66 | 34.50 | 27.08 | 24.50 | 23.44 | 23.69 | 22.56 |
| Max Improve % | | - | - | - | - | - | 95.0 | 122.4 | 181.6 | 154.7 | 277.0 | 184.5 | 187.0 | 179.5 |

within the FedGNN framework. To demonstrate the generalizability of CC-MIA, results using GAT and GraphSAGE are presented in Appendix A.12. We further assess CC-MIA under five widely adopted federated approaches: FedAvg McMahan et al. (2017), FedProx Li et al. (2020), SCAFFOLD Karimireddy et al. (2020), FedDF Lin et al. (2020), and FedNova Wang et al. (2020). Detailed descriptions can be found in Appendix A.7.

The evaluation is conducted on six benchmark datasets for node classification: Cora Yang et al. (2016), Citeseer Yang et al. (2016), PubMed Yang et al. (2016), CS, Physics Shchur et al. (2018), and DBLP Bojchevski & Günnemann (2018), with dataset details summarized inAppendix A.10. For comparison, we include three strong MIA baselines: HP-MIA Chen et al. (2024), GAN-Based Data Enhancement Sui et al. (2023), and CS-MIA Gu et al. (2022), described in Appendix A.8. Lastly, the computational cost of CC-MIA and explore potential defense strategies in Appendix A.14 and Appendix A.15, respectively.

## 5.2 EXPERIMENTAL SETTINGS

In our experiments, the target dataset is partitioned into $k$ subgraphs using METIS Karypis & Kumar (1998), with one subgraph assigned to each client, leveraging its efficiency, balanced non-overlapping partitions, and deterministic output for reliable and reproducible initialization Wang et al. (2025); Liang et al. (2025); Chen et al. (2025); Ma et al. (2025). Due to the graph structure, nodes of similar classes are clustered closely within subgraphs, leading to a non-i.i.d. distribution of nodes across clients. A statistical analysis of the node class distributions for each client is provided in Fig 4. Specifically, 40% of the target dataset is allocated as the training-set, while the entire shadow dataset is used for the training-set MI attack model. Both the GNNs and the attacker models are trained with a learning rate of $1e-3$. The number of hidden neurons in the GNN and the attack model is set to 128. For the hyperparameters in Eq. 12, $\alpha$ and $\beta$ are set to $1e-3$ and $1e-4$, respectively. We evaluate membership inference and client-data-ownership using **AUC** and **Accuracy**, respectively, while **AUC** and **RNMSE** assess gradient inversion quality. Details are in Appendix A.9.

## 5.3 MEMBERSHIP INFERENCE ATTACK RESULTS

To evaluate the attack performance, we first initialize CC-MIA on the GCN as the FL global model. Specifically, we report the shadow dataset for each target dataset that achieves the best attack performance when the number of clients is set to 5. **AUC** is employed to evaluate attack performance, ensuring that biases introduced by training-set proportion are effectively mitigated. The MIA results obtained using the optimal shadow data set are reported on the left side of Table 1. The proposed CC-MIA consistently outperforms all MIA baselines across different federated settings. On Citeseer with SCAFFOLD, it achieves up to 72.16% improvement. In contrast, HP-MIA and GAN-based methods, designed for centralized models, perform only slightly above random guessing (AUCs of 56.09% and 58.14% on PubMed and Citeseer, respectively). While CS-MIA performs better in

federated settings, it still falls short due to its lack of GNN-specific design. Note that the superior performance of `CC-MIA` with Physics as the shadow dataset under SCAFFOLD stems from its correction mechanism, enhancing alignment between Physics and PubMed for better inference.

More results on other datasets (CS, Physics, and DBLP) are reported in Appendix Table 5. For generalization purposes, we also proposed the MIA attack results on GAT and GraphSAGE in Table 7 and 8, respectively.

### 5.4 CLIENT-DATA IDENTIFICATION RESULTS

We evaluate the performance of `CC-MIA`'s client-data identification attack within the federated GCNs. Specifically, we test with the number of clients ranging from 3 to 10. A larger number of clients increases the difficulty of achieving a successful attack. For prototype computation, we use node features extracted from the 1-st GCN layer of the trained global GNN, the results are presented on the right side of Table 1. For comparison, we provide the baseline probability for $k$-clients, referred to as **Client-uniform Probability**. For the convenience of comparison, we attach the probabilities of random selection. The optimal performance of `CC-MIA` across different clients consistently exceeds client-uniform probability by at least 95%. Notably, the classification accuracy shows a significant improvement as the number of clients increases, particularly beyond 7 clients, highlighting the method's scalability in more complex federated settings. More datasets can be referred to Table 6.

### 5.5 ABLATION STUDIES AND VARIANTS

We conducted ablation studies using GCN to evaluate the impact of `CC-MIA` components. As shown in Table 2, removing the shadow dataset (`CC-MIA` *no shadow*) lowers AUC to below 60% on Cora, PubMed, and Physics, though it still outperforms baselines. Using subgraph-based shadow datasets further reduces performance, as full shadow datasets can better capture data distributions.

For client data ownership identification, removing the prototype strategy (`CC-MIA` *no prot*) weakens class differentiation, reducing AUC from 27.95% to 20.82% for 10 clients. The `CC-MIA` (*norm*) variant performs near random (AUC 9.12%), emphasizing cosine similarity's effectiveness in high-dimensional feature alignment. The details of the analysis of the variants implementation on `CC-MIA` (subgraph) are discussed in Appendix A.11.

Table 2: Ablations and variants of member inference (left) and client-data identification (right).

| | Member inference | | | | | | | Client-data identification | | | | |
|---|---|---|---|---|---|---|---|---|---|---|---|---|
| Dataset | Cora | Citeseer | PubMed | CS | Physics | DBLP | | Client Num | 3 | 5 | 7 | 10 |
| CC-MIA | **82.04** | **86.04** | **71.73** | **81.60** | **73.00** | **83.42** | | CC-MIA | **59.27** | **56.31** | **34.08** | **27.95** |
| CC-MIA (*no shadow*) | 57.86 | 67.01 | 59.57 | 64.66 | 57.58 | 61.63 | | CC-MIA (*no prot*) | 51.99 | 43.46 | 31.68 | 20.82 |
| CC-MIA (*subgraph*) | 62.92 | 79.69 | 67.76 | 76.62 | 70.89 | 78.09 | | CC-MIA (*norm*) | 35.16 | 18.76 | 12.90 | 9.12 |

### 5.6 INVERSION QUALITY

We measure the fidelity of gradient-inverted subgraphs using edge-AUC (higher is better) and feature-RNMSE (lower is better) from Zhang et al. (2021b); Anand Sinha et al. (2024) (Appendix A.9). Table 3 shows reconstruction quality declining as client count increases—for example, PubMed's adjacency AUC falls from 65.98% (3 clients) to 60.87% (10 clients), while RNMSE rises from 0.0018 to 0.0030. Larger, more complex graphs like PubMed and Physics maintain relatively low RNMSE even with many clients, whereas small graphs (*e.g.*, CS) achieve the highest AUC (79.07%) and lowest RNMSE (0.0002) under three-client settings. Fig 2 visualizes reconstructed versus true subgraph embeddings on Citeseer (4 clients) using t-SNE on first-layer GCN features. Reconstructed clusters are more compact, class-consistent, and often exhibit greater linear separability. This reduces ambiguity in dense regions and enhances the preservation of class-specific structures.

### 5.7 PROTOTYPE VISUALIZATION

To demonstrate `CC-MIA`'s effectiveness in client-data identification attacks, we visualize prototypes for different clients using t-SNE on the Cora dataset based on GCN, reducing prototype dimensions

Table 3: Performance of `CC-MIA` in the gradient inversion of the federated GCN

| Dataset | | 3-Client | 5-Client | 7-Client | 10-Client | | Dataset | | 3-Client | 5-Client | 7-Client | 10-Client |
|---|---|---|---|---|---|---|---|---|---|---|---|---|
| Cora | AUC ↑ | 76.52 | 69.78 | 68.48 | 69.59 | | CS | AUC ↑ | 79.07 | 72.71 | 72.44 | 67.72 |
| | RNMSE ↓ | 0.0027 | 0.0028 | 0.0041 | 0.0044 | | | RNMSE ↓ | 0.0002 | 0.0003 | 0.0003 | 0.0005 |
| Citeseer | AUC ↑ | 76.09 | 74.23 | 74.78 | 64.89 | | Physics | AUC ↑ | 68.67 | 65.43 | 62.34 | 62.53 |
| | RNMSE ↓ | 0.0017 | 0.0021 | 0.0024 | 0.0026 | | | RNMSE ↓ | 0.0002 | 0.0002 | 0.0002 | 0.0003 |
| PubMed | AUC ↑ | 65.98 | 62.67 | 61.22 | 60.87 | | DBLP | AUC ↑ | 69.52 | 65.12 | 64.48 | 65.01 |
| | RNMSE ↓ | 0.0018 | 0.0022 | 0.0021 | 0.0030 | | | RNMSE ↓ | 0.0013 | 0.0017 | 0.0019 | 0.0019 |

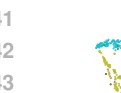 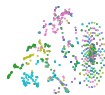 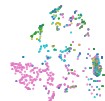 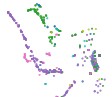 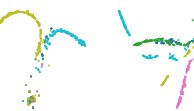 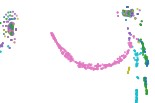 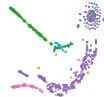

(a) Clt1-GT  (b) Clt2-GT  (c) Clt3-GT  (d) Clt4-GT  (e) Clt1-Inv  (f) Clt2-Inv  (g) Clt3-Inv  (h) Clt4-Inv

Figure 2: Inverse comparison of node features of each client on Citeseer. Clt: Client; GT: Ground-truth; Inv: Inverse.

to 2D (Fig 3). Prototypes from the same client share the same shape, while those from the same class share the same color. Results with 3–8 clients show that prototypes from the same client cluster tightly, with separability improving as client numbers decrease, simplifying classification. These trends align with results in Section 5.4. `CC-MIA` 's prototypes exhibit high discriminability, enabling reliable client identification by comparing the queried node's position relative to client prototypes.

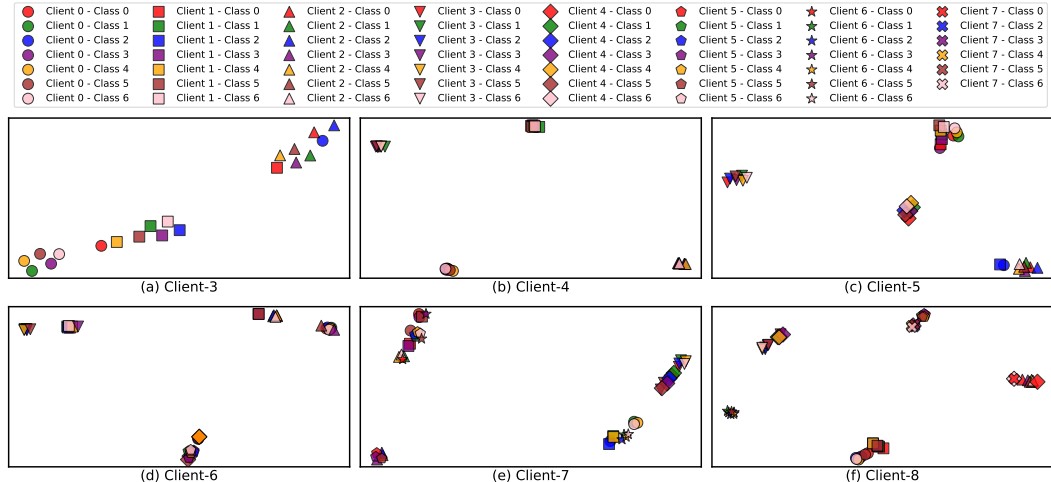

Figure 3: The visualization of each client prototype on Cora generated by `CC-MIA`.

# 6 CONCLUSION

This work sheds light on a previously underexplored privacy risk in federated GNNs: cross-client membership inference attacks. We propose `CC-MIA`, a novel attack framework that goes beyond traditional sample-level inference by identifying the client origin of individual data samples. By exploiting structural patterns, gradient dynamics, and embedding behaviors during training, `CC-MIA` effectively links nodes to their source clients. Experiments across diverse graph datasets confirm the attack's effectiveness under realistic FL settings. These findings emphasize the urgency of developing privacy-preserving mechanisms that are robust not only to data-level inference but also to client-level attribution threats in FedGNNs. The **limitations** of `CC-MIA` are discussed in Appendix A.6.

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

# A Appendix

## A.1 The Use of Large Language Models

The language of this paper was polished using large language models (LLMs) to enhance clarity and readability. The final content and academic integrity remain the responsibility of the authors.

## A.2 Motivation

Most existing GNN-specific attacks are confined to centralized settings, where the assumption of full graph access significantly simplifies the attacker's task. However, the advent of federated GNNs introduces enhanced privacy protection mechanisms that render traditional centralized GNN MIA methods ineffective. This limitation arises because federated frameworks distribute disjoint and heterogeneous subgraphs among clients. Attackers, treated as equal participants within this framework, face significant constraints: they are unable to access similar shadow subgraphs from other clients and lack the posterior probabilities of other clients' GNN models, which are critical for training conventional MIA classifiers. These challenges give rise to the first question: *Is it feasible to*

*perform cross-client membership inference without any knowledge of the GNN models employed by other clients and without access to their subgraphs?*

While prior research on MIAs in federated learning has primarily focused on sample-level membership inference within training datasets, the specific problem of node-level ownership inference in federated GNNs remains underexplored. Node-level MIAs differ fundamentally from graph-level attacks, which bear similarities to traditional sample-level attacks (*e.g.*, those targeting images or text). The granular nature of nodes as integral components of a graph introduces unique complexities not present in graph-level analyses.

As detailed in Appendix A.10, the subgraph structures held by individual clients in federated GNNs are typically non-i.i.d., reflecting the inherent heterogeneity induced by graph partitioning algorithms. These algorithms often result in imbalanced class distributions across clients Xie et al. (2021a). This observation motivates the second research question: *Can an attacker, operating as an equal client within the federated framework, perform node-level client ownership inference?*

We have solved these problems with our proposed `CC-MIA` in Section 4

### A.3 ALGORITHMS

We present the algorithmic workflows for Gradient Inversion and Prototype-Based Client Matching in Subsection 4.2, detailed in Algorithm 1 and Algorithm 2, respectively.

---

**Algorithm 1:** Workflow of Gradient Inversion.

---

**Input:** Global GNN $\mathcal{F}(X, A; W)$; Intercepted client gradients $\nabla_W \mathcal{L}$; Ground-truth labels $Y$.

**Output:** Reconstructed client data $(\hat{X}, \hat{A})$.

1 Initialize dummy node features $\hat{X}^{(1)}$ and adjacency matrix $\hat{A}^{(1)}$;

2 **for** *epoch* $e = 1$ **to** $T$ **do**

3      Compute synthetic gradient: $\nabla_W \hat{\mathcal{L}} = \partial \mathcal{L}(F(\hat{X}^e, \hat{A}^e, W), Y)/\partial W$;

4      Evaluate gradient-fitting loss $\hat{\mathbb{L}}^{(e)}$ (Eq. 12);

5      Gradient descent for dummy node features: $\hat{X}^{e+1} = \hat{X}^e - \eta_x \nabla_{\hat{X}^e} \hat{\mathbb{L}}^e$;

6      Gradient descent for dummy adjacency: $\hat{A}^{e+1} = \text{proj}_{[0,1]}(\hat{A}^e - \eta \nabla_{\hat{A}^e} \hat{\mathbb{L}}^e)$

7 **end**

8 Obtain $\hat{A}$ by sampling $\hat{A}^{e+1}$ (Eq. 14);

9 **return** $\hat{X} = \hat{X}^{e+1}, \hat{A}$

---

**Algorithm 2:** Workflow of Prototype-based Client Matching

---

**Input:** Global GNN model $\mathcal{F}(X, A; W)$, reconstructed subgraphs $\{(\hat{X}_k, \hat{A}_k)\}_{k=1}^{K}$, real graph $(X, A)$

**Output:** Client assignment $\{\hat{k}_i\}_{i=1}^{N}$ for each node $x_i \in X$

1 **for** $k \leftarrow 1$ **to** $K$ **do**

2      $\hat{\mathcal{E}}_k \leftarrow \mathcal{F}_1(\hat{X}_k, \hat{A}_k)$;

3      **for** *each class* $c \in \mathcal{C}_k$ **do**

4          $\mu_c^{(k)} \leftarrow \frac{1}{|\mathcal{I}_c^k|} \sum_{j \in \mathcal{I}_c^k} \hat{\mathcal{E}}_k^j$

5      **end**

6      $\mathcal{P}_k \leftarrow \{\mu_c^{(k)} \mid c \in \mathcal{C}_k\}$    // Eq. 15

7 **end**

8 $\mathcal{E} \leftarrow \mathcal{F}_1(X, A)$;

9 **for** $i \leftarrow 1$ **to** $N$ **do**

10      **for** $k \leftarrow 1$ **to** $K$ **do**

11          **if** $y_i \in \mathcal{C}_k$ **then**

12              $d_{i,k} \leftarrow 1 - \frac{\mathcal{E}_i \cdot \mu_{y_i}^{(k)}}{\|\mathcal{E}_i\| \, \|\mu_{y_i}^{(k)}\|}$    // Eq. 16;

13          **else**

14              $d_{i,k} \leftarrow \infty$;

15          **end**

16      **end**

17      $\hat{k}_i \leftarrow \arg\min_k d_{i,k}$

18 **end**

19 **return** $\{\hat{k}_i\}_{i=1}^{N}$

---

### A.4 COMPLEXITY

We analyse the time complexity and space complexity of the membership inference and client-data identification tasks of `CC-MIA`, respectively.

#### A.4.1 MEMBERSHIP INFERENCE

The complexity of membership inference primarily arises from two stages: global GNN inference and MLP classification.

For an $L$-layer GNN with $N$ nodes and feature dimensions of $D$, the feature transformation and neighborhood aggregation steps have a time complexity of $O\left(L(ND^2 + |E|D)\right)$. Memory usage includes storing the adjacency matrix ($O(|E|)$), node embeddings ($O(ND)$), and model weights ($O(LD^2)$), resulting in a space complexity of $O\left(|E| + ND + LD^2\right)$.

For a $J$-layer MLP with width $O(D)$, the computation cost per node is $O(JD^2)$. Processing $N$ nodes leads to a total time complexity of $O(NJD^2)$. The space complexity is dominated by storing parameters, requiring $O(JD^2)$ memory.

Overall, membership inference is dominated by the GNN inference. For large graphs ($|E| \gg N$), the sparse aggregation $O(L|E|D)$ becomes the primary bottleneck. If the embedding dimension $D$ is large, the dense transformation term $O(LND^2)$ also contributes significantly. The MLP, with typically small $J$, incurs relatively minor costs. Thus, the overall time and space complexities are approximately $\mathbf{O}\left(\mathbf{L}(\mathbf{ND^2} + |\mathbf{E}|\mathbf{D})\right)$ and $\mathbf{O}\left(|\mathbf{E}| + \mathbf{ND} + \mathbf{LD^2}\right)$.

#### A.4.2 CLIENT-DATA IDENTIFICATION

The main computational overhead of client-data identification lies in gradient inversion and GNN feature extraction. Each optimization iteration involves a forward and backward pass on an $n$-node subgraph with an $L$-layer GNN. The time complexity for a single layer is $O(N \cdot D^2 + |E| \cdot D)$, where $N$ is the number of nodes, $D$ is the embedding dimension, and $|E|$ is the number of edges. For $T$ steps of gradient descent, the total time complexity of gradient inversion is approximately: $O(T \cdot L \cdot (N \cdot D^2 + |E| \cdot D)) \approx O(T \cdot L \cdot N \cdot D^2)$, assuming $|E| = O(N)$ for sparse graphs. Memory consumption in this stage is dominated by storing the reconstructed inputs, which include the adjacency matrix ($O(N^2)$) and node features ($O(N \cdot D)$), resulting in a total space complexity of $O(N^2 + N \cdot D)$.

The feature extraction involves a single forward pass of the global GNN on the reconstructed subgraph. The time complexity is: $O(L \cdot (N \cdot D^2 + |E| \cdot D)) \approx O(L \cdot N \cdot D^2)$, with space complexity of $O(N \cdot D + |E|)$.

The prototype matching is the least computationally intensive. The $D$-dimensional embedding of the target node is compared with $K \cdot C$ prototypes, where $K$ is the number of clients and $C$ is the number of classes. The time and space complexity of this stage are both $O(K \cdot C \cdot D)$.

To summarize, the overall complexity of client-data identification is primarily determined by gradient inversion, with a time complexity of $\mathbf{O}(\mathbf{T} \cdot \mathbf{L} \cdot \mathbf{N} \cdot \mathbf{D^2})$ and a space complexity of $\mathbf{O}(\mathbf{N^2} + \mathbf{N} \cdot \mathbf{D})$.

### A.5 THEORETICAL ANALYSIS OF ATTACKS

#### A.5.1 MEMBERSHIP INFERENCE ATTACK (MIA)

**Objective.** Given shadow data, the adversary trains a binary classifier $f_w$ to determine whether a target node $x_i$ belongs to a client's training set.

**Key factors.** The effectiveness of MIA depends on (i) the similarity between shadow and target distributions, (ii) the shadow sample size $n$, and (iii) the classifier's complexity.

**Proposition 1 (PAC-style bound)** *Assume that the embedding distributions of training members and non-members are denoted as $P_{mem}$ and $P_{non}$, with divergence*

$$d(P_{mem}, P_{non}) \geq \tau > 0. \tag{17}$$

*Suppose the adversary trains a classifier of complexity $V$ on $n$ shadow samples, achieving empirical error $err_{emp}$. Then, with probability at least $1 - \delta$,*

$$err_{true} \leq err_{emp} + C\sqrt{\frac{V + \log(1/\delta)}{n}}. \tag{18}$$

**Proof A.1** *By standard PAC/Rademacher complexity analysis, the true risk of the shadow-trained classifier is bounded by the empirical risk plus a complexity term. Since $d(P_{mem}, P_{non}) \geq \tau$, the Bayes optimal error is strictly below 0.5, ensuring that with sufficiently large $n$ and small empirical error, the classifier generalizes and achieves an AUC significantly above random guessing.*

**Proposition 2 (Impact of shadow similarity)** *Let $S(G_{shadow}, G_{target})$ denote a structural similarity measure, where smaller values indicate higher similarity. Then, smaller $S$ leads to higher attack success, while large $S$ degrades performance towards random guessing.*

### A.5.2 CLIENT-DATA IDENTIFICATION

**Procedure.** The adversary intercepts client gradients and performs gradient inversion to reconstruct data $(\widehat{X}, \widehat{A})$. Node embeddings are then extracted to build class prototypes $\widehat{\mu}_c^{(k)}$, which are matched via cosine similarity.

**Reconstruction loss.**

$$L = 1 - \frac{\langle \nabla_W L, \, \nabla_W \widehat{L} \rangle}{\|\nabla_W L\|\|\nabla_W \widehat{L}\|}, \quad \widehat{L} = L + \alpha L_{\text{smooth}} + \beta\|\widehat{A}\|_F^2. \tag{19}$$

**Proposition 3 (Margin-based guarantee)** *Suppose the reconstruction perturbation satisfies $\|e_i\| \leq \varepsilon$ for embeddings and $\|\delta_c^{(k)}\| \leq \varepsilon_p$ for prototypes. Assume embeddings and prototypes have norms lower bounded by $m > 0$. For a node $i$ belonging to client $k^\star$ with class $y_i$, define the cosine margin*

$$\Delta := \cos(E_i, \mu_{y_i}^{(k^\star)}) - \max_{k \neq k^\star} \cos(E_i, \mu_{y_i}^{(k)}). \tag{20}$$

*If*

$$\Delta > \frac{2(\varepsilon + \varepsilon_p)}{m}, \tag{21}$$

*then prototype matching via cosine similarity remains correct, i.e., the node is assigned to the true client $k^\star$.*

**Proof A.2** *Cosine similarity is Lipschitz continuous under perturbations. With bounded perturbations $\varepsilon$ and $\varepsilon_p$, the variation in cosine similarity is at most $(\varepsilon + \varepsilon_p)/m$. If the true margin $\Delta$ exceeds twice this bound, the index of the maximum cosine similarity is unchanged, ensuring correct client assignment.*

**Influencing factors.** The guarantee highlights the role of (i) the eavesdropping probability $\gamma$, (ii) the inversion quality controlled by iteration steps $T$ and regularization $\alpha, \beta$, and (iii) the number of clients $K$, since larger $K$ reduces margins and makes identification more difficult.

### A.6 LIMITATIONS

The **limitations** of CC-MIA include the degradation of gradient inversion quality as the number of clients increases, indicating scalability issues in reconstructing accurate subgraph structures under large-scale federated settings. Moreover, CC-MIA assumes class-consistent gradients and prototype separability for effective ownership inference, which may not hold under real-world non-IID client distributions.

### A.7 FEDERATED LEARNING APPROACHES

We evaluate the proposed CC-MIA under various federated approaches to demonstrate its generalizability in federated GNN scenarios. The specific methods include:

- **FedAvg** McMahan et al. (2017) is the foundational algorithm for federated learning, where clients perform local training, and the server aggregates model parameters via weighted averaging. The weights are typically proportional to the data volume of each client relative to the total dataset.
- **FedProx** Li et al. (2020) is an extension of FedAvg that introduces a regularization term in the client loss function to enforce consistency with the global model, thereby aligning local updates with the global objective and accelerating convergence.
- **SCAFFOLD** Karimireddy et al. (2020) incorporates control variates to correct gradient deviations caused by data heterogeneity. By coordinating global and local control variates, it mitigates client drift and improves convergence in non-IID settings.
- **FedDF** Lin et al. (2020) is a robust distillation-based framework for federated model fusion. It supports heterogeneity in client models, data distributions, and neural architectures, providing flexibility and resilience in federated learning setups.
- **FedNova** Wang et al. (2020) is an enhancement of FedAvg that dynamically adjusts aggregation weights by considering client-specific local iteration counts and data volumes. This approach addresses the challenges posed by heterogeneous data and varying local training steps.

## A.8    BASELINES

We set the following strong baselines in the membership inference.

- **HP-MIA** Chen et al. (2024) trains the membership inference model by comparing the differences in feature extraction between the server and the attacker using the data owned by the attacker. It leverages the phenomenon of model overfitting to distinguish training-set data from non-members.
- **GAN Based Data Enhancement** Sui et al. (2023) utilizes generative adversarial networks (GANs) to augment data. By generating additional synthetic data that aligns with the attacker's data distribution, it enhances the training dataset used to train the membership inference model.
- **CS-MIA** Gu et al. (2022) is a membership inference attack method designed for federated learning settings. It employs the confidence sequences from the model at each round on the attacker's local data to train the membership inference model.

## A.9    METRICS

We explain the evaluation metrics used in this study. For membership inferences, AUC mitigates bias from varying training-set proportions. Client-data identification uses accuracy for classification of evenly divided nodes. Subgraph reconstruction via gradient inversion is evaluated using AUC for graph structure and RNMSE for node features.

- **AUC (Area Under Curve)**: The area under the Reciever Operator Characteristic Curve.
- The **accuracy** of the client-data identificationis formalized as:

$$ACC_{\text{Client-data Identification}} = \frac{\sum_{i=1}^{N}(\hat{k}_i == k_i)}{N}, \tag{22}$$

where $k_i$ denotes the true client that node $i$ belongs to.

- The **Root Normalised Mean Squared Error(RNMSE)** is formalized as:

$$RNMSE(x_v, \hat{x}_v) = \frac{\|x_v - \hat{x}_v\|}{\|x_v\|}. \tag{23}$$

## A.10    DATA STATISTICS

We report the statistics of the datasets we used, as shown in Table 4. For instance, we partition the target dataset into subgraphs for 5 clients using the METISKarypis & Kumar (1998). The class

Table 4: Dataset Statistics

| Dataset | Cora | Citeseer | PubMed | CS | Physics | DBLP |
|---------|------|----------|--------|-----|---------|------|
| $|V|$ | 2,708 | 3,327 | 19,717 | 18,333 | 34,493 | 17,716 |
| $|E|$ | 10,556 | 9,104 | 88,648 | 163,788 | 495,924 | 105,734 |
| # Classes | 7 | 6 | 3 | 15 | 5 | 4 |

(a) Cora-Client1    (b) Cora-Client2    (c) Cora-Client3    (d) Cora-Client4    (e) Cora-Client5

(f) Citeseer-Client1   (g) Citeseer-Client2   (h) Citeseer-Client3   (i) Citeseer-Client4   (j) Citeseer-Client5

(k) PubMed-Client1   (l) PubMed-Client2   (m) PubMed-Client3   (n) PubMed-Client4   (o) PubMed-Client5

Figure 4: Class distribution in different clients based on citation networks.

distribution of nodes for each client is illustrated in Fig 4. All evaluated membership inference attacks (MIAs) are conducted in heterogeneous scenarios where the data distribution is inherently unbalanced between clients.

## A.11 VARIENTS IMPLEMENTATION

**CC-MIA (*subgraph*)** To quantify the structural similarity between $G_{\text{inv}}$ and $G_{\text{shadow}}$, we compute the degree and clustering coefficient distributions of their nodes. Let $\deg_G(v)$ represent the degree of node $v$ in graph $G$. The degree distributions $p_1(k)$ and $p_2(k)$ are defined as:

$$p_1(k) = \frac{\left|\{v \in V_{\text{inv}} : \deg_{G_{\text{inv}}}(v) = k\}\right|}{|V_{\text{inv}}|}, \quad p_2(k) = \frac{\left|\{w \in V_{\text{shadow}} : \deg_{G_{\text{shadow}}}(w) = k\}\right|}{|V_{\text{shadow}}|}. \tag{24}$$

Similarly, the clustering coefficient distributions $q_1(j)$ and $q_2(j)$ are computed by dividing the clustering coefficient range $[0, 1]$ into bins $B_j$:

$$q_1(j) = \frac{\left|\{v \in V_{\text{inv}} : \text{CC}_{G_{\text{inv}}}(v) \in B_j\}\right|}{|V_{\text{inv}}|}, \quad q_2(j) = \frac{\left|\{w \in V_{\text{shadow}} : \text{CC}_{G_{\text{shadow}}}(w) \in B_j\}\right|}{|V_{\text{shadow}}|}. \tag{25}$$

Table 5: Performance (AUC %) of federated GCN member inference attacks on other datasets.

| Dataset | Shadow Dataset | FL Approaches | HP-MIA | GAN-Based | CS-MIA | CC-MIA |
|---------|----------------|---------------|--------|-----------|--------|--------|
| CS | Cora | FedAvg | 53.46 | 50.47 | 68.24 | **81.60** |
| | Citeseer | FedProx | 52.83 | 50.84 | 59.78 | **64.70** |
| | DBLP | SCAFFOLD | 55.87 | 53.01 | 58.28 | **72.46** |
| | Citeseer | FedDF | 50.13 | 55.34 | 61.47 | **78.82** |
| | DBLP | FedNova | 53.42 | 56.12 | 58.44 | **76.27** |
| Physics | PubMed | FedAvg | 54.67 | 57.04 | 59.15 | **73.00** |
| | Citeseer | FedProx | 55.21 | 53.89 | 79.21 | **82.45** |
| | DBLP | SCAFFOLD | 56.09 | 59.78 | 61.54 | **80.12** |
| | Citeseer | FedDF | 57.73 | 58.56 | 73.75 | **79.88** |
| | PubMed | FedNova | 56.92 | 57.45 | 52.63 | **75.99** |
| DBLP | PubMed | FedAvg | 53.47 | 54.98 | 61.91 | **83.42** |
| | PubMed | FedProx | 54.74 | 53.21 | 54.74 | **81.75** |
| | PubMed | SCAFFOLD | 55.99 | 52.08 | 58.23 | **85.73** |
| | PubMed | FedDF | 50.61 | 55.32 | 60.40 | **78.50** |
| | PubMed | FedNova | 51.82 | 54.45 | 55.65 | **85.22** |

Table 6: Performance of federated GCN client-data identification on other datasets. Clts: Clients.

| Dataset | FL Approaches | 3-Clts | 4-Clts | 5-Clts | 6-Clts | 7-Clts | 8-Clts | 9-Clts | 10-Clts |
|---------|---------------|--------|--------|--------|--------|--------|--------|--------|---------|
| | Random | **33.33** | **25.00** | **20.00** | **16.66** | **14.29** | **12.50** | **11.11** | **10.00** |
| CS | FedAvg | 41.78 | 36.78 | 34.81 | 29.15 | 27.49 | 26.32 | 24.79 | 21.63 |
| | FedProx | 38.26 | 34.35 | 30.16 | 30.11 | 27.92 | 23.37 | 19.66 | 19.19 |
| | SCAFFOLD | 52.60 | 40.58 | 37.32 | 32.10 | 29.01 | 26.82 | 18.39 | 17.66 |
| | FedDF | 48.07 | 45.57 | 37.42 | 33.40 | 28.53 | 26.05 | 25.14 | 22.98 |
| | FedNova | 50.45 | 34.79 | 32.14 | 30.12 | 29.11 | 26.32 | 23.37 | 20.51 |
| Physics | FedAvg | 42.75 | 38.40 | 34.05 | 30.25 | 28.15 | 27.63 | 27.46 | 20.43 |
| | FedProx | 48.25 | 42.14 | 37.89 | 31.04 | 26.03 | 24.09 | 22.54 | 19.15 |
| | SCAFFOLD | 45.15 | 40.60 | 35.10 | 31.45 | 29.02 | 25.56 | 23.03 | 20.57 |
| | FedDF | 40.10 | 36.42 | 32.04 | 28.00 | 24.54 | 20.36 | 18.15 | 15.74 |
| | FedNova | 41.05 | 30.78 | 29.62 | 26.34 | 24.36 | 20.66 | 20.47 | 18.95 |
| DBLP | FedAvg | 53.13 | 43.35 | 37.38 | 32.87 | 31.76 | 30.80 | 30.10 | 22.92 |
| | FedProx | 72.25 | 44.42 | 42.26 | 34.94 | 34.19 | 31.99 | 28.56 | 28.77 |
| | SCAFFOLD | 43.13 | 39.13 | 37.69 | 32.24 | 26.21 | 25.82 | 20.61 | 18.43 |
| | FedDF | 60.58 | 46.87 | 38.48 | 37.79 | 30.17 | 27.02 | 21.53 | 20.94 |
| | FedNova | 36.20 | 31.27 | 25.06 | 22.00 | 20.15 | 18.43 | 16.11 | 14.15 |

To stabilize the calculation of KL divergence, we truncate probabilities below a threshold $\epsilon = 10^{-3}$:

$$p'_{1,k} = \max\big(p_1(k), \epsilon\big), \quad p'_{2,k} = \max\big(p_2(k), \epsilon\big), \quad q'_{1,j} = \max\big(q_1(j), \epsilon\big), \quad q'_{2,j} = \max\big(q_2(j), \epsilon\big). \tag{26}$$

The KL divergence for the degree and clustering coefficient distributions is then:

$$D_{\mathrm{KL}}(p_1\|p_2) = \sum_k p'_{1,k} \ln \frac{p'_{1,k}}{p'_{2,k}}, \quad D_{\mathrm{KL}}(q_1\|q_2) = \sum_j q'_{1,j} \ln \frac{q'_{1,j}}{q'_{2,j}}. \tag{27}$$

The structural similarity score $S$ between the two subgraphs is:

$$S = D_{\mathrm{KL}}(p_1\|p_2) + D_{\mathrm{KL}}(q_1\|q_2) = \sum_k p'_{1,k} \ln \frac{p'_{1,k}}{p'_{2,k}} + \sum_j q'_{1,j} \ln \frac{q'_{1,j}}{q'_{2,j}}. \tag{28}$$

## A.12 ADDITIONAL PERFORMANCE OF CC-MIA

We report the performance of additional datasets under federated GCN, with results for membership inferences and client-data identification detailed in Table 5 and Table 6, respectively. Furthermore, we report the membership inference performance of federated GAT and federated GraphSAGE across all datasets, as shown in Table 7 and Table 8.

## A.13 CONVERGENCE ANALYSIS

To validate the effectiveness of CC-MIA, we conducted convergence evaluations for both membership inference and client-data identification. Specifically, for membership inference, we plot the loss

Table 7: Performance (AUC %) of federated GAT Velickovic et al. (2018) member inference.

| Dataset | Shadow Dataset | FL Approaches | HP-MIA | GAN-Based | CS-MIA | CC-MIA |
|---|---|---|---|---|---|---|
| Cora | Citeseer | FedAvg | 56.32 | 54.87 | 58.12 | **65.36** |
| | CS | FedProx | 50.14 | 49.98 | 59.67 | **60.41** |
| | CS | SCAFFOLD | 52.73 | 51.44 | 57.21 | **64.32** |
| | PubMed | FedDF | 53.98 | 52.10 | 58.79 | **62.96** |
| | DBLP | FedNova | 54.57 | 56.02 | 54.31 | **61.77** |
| Citeseer | PubMed | FedAvg | 51.25 | 50.34 | 53.88 | **64.02** |
| | DBLP | FedProx | 52.90 | 53.12 | 54.05 | **61.75** |
| | DBLP | SCAFFOLD | 49.68 | 50.77 | 51.29 | **58.72** |
| | DBLP | FedDF | 54.01 | 51.93 | 54.42 | **60.13** |
| | DBLP | FedNova | 53.77 | 58.00 | 57.84 | **62.99** |
| PubMed | DBLP | FedAvg | 55.12 | 50.68 | 56.45 | **63.00** |
| | DBLP | FedProx | 54.40 | 54.37 | 55.18 | **63.93** |
| | DBLP | SCAFFOLD | 53.21 | 54.52 | 49.87 | **62.33** |
| | DBLP | FedDF | 56.00 | 54.66 | 57.39 | **62.30** |
| | DBLP | FedNova | 55.93 | 52.89 | 58.71 | **63.32** |
| CS | Cora | FedAvg | 53.01 | 49.12 | 55.90 | **62.08** |
| | Citeseer | FedProx | 52.48 | 50.57 | 53.33 | **62.87** |
| | Citeseer | SCAFFOLD | 55.04 | 52.88 | 54.17 | **61.97** |
| | Cora | FedDF | 50.88 | 55.01 | 52.74 | **61.96** |
| | Citeseer | FedNova | 53.19 | 56.00 | 51.62 | **60.50** |
| Physics | DBLP | FedAvg | 54.29 | 57.33 | 54.88 | **63.38** |
| | DBLP | FedProx | 55.95 | 53.60 | 56.77 | **62.23** |
| | DBLP | SCAFFOLD | 56.47 | 59.01 | 52.18 | **61.52** |
| | CS | FedDF | 57.11 | 58.02 | 54.01 | **61.73** |
| | DBLP | FedNova | 56.58 | 57.20 | 50.97 | **62.36** |
| DBLP | Cora | FedAvg | 53.84 | 54.33 | 52.10 | **60.21** |
| | Citeseer | FedProx | 54.09 | 53.68 | 54.76 | **63.85** |
| | Citeseer | SCAFFOLD | 55.23 | 52.46 | 56.49 | **63.57** |
| | Citeseer | FedDF | 50.47 | 55.18 | 53.82 | **63.27** |
| | Citeseer | FedNova | 51.68 | 54.72 | 55.09 | **64.28** |

Table 8: Performance (AUC %) of federated GraphSAGE Hamilton et al. (2017) member inference.

| Dataset | Shadow Dataset | FL Approaches | HP-MIA | GAN-Based | CS-MIA | CC-MIA |
|---|---|---|---|---|---|---|
| Cora | DBLP | FedAvg | 55.12 | 53.78 | 57.45 | **65.82** |
| | PubMed | FedProx | 50.97 | 49.35 | 58.76 | **71.47** |
| | DBLP | SCAFFOLD | 52.43 | 51.21 | 56.67 | **61.35** |
| | DBLP | FedDF | 53.84 | 52.98 | 58.47 | **63.38** |
| | PubMed | FedNova | 53.76 | 55.32 | 53.94 | **72.17** |
| Citeseer | PubMed | FedAvg | 50.81 | 50.16 | 54.02 | **71.26** |
| | PubMed | FedProx | 52.95 | 52.41 | 53.18 | **67.83** |
| | PubMed | SCAFFOLD | 48.73 | 49.89 | 49.67 | **63.35** |
| | DBLP | FedDF | 54.61 | 51.65 | 53.14 | **63.60** |
| | PubMed | FedNova | 54.02 | 56.11 | 56.72 | **64.91** |
| PubMed | Cora | FedAvg | 53.84 | 50.37 | 56.12 | **62.37** |
| | Cora | FedProx | 54.04 | 53.18 | 55.24 | **60.57** |
| | Cora | SCAFFOLD | 52.98 | 53.76 | 50.14 | **61.98** |
| | Cora | FedDF | 55.71 | 54.11 | 56.72 | **61.64** |
| | Cora | FedNova | 54.89 | 53.24 | 57.94 | **60.99** |
| CS | DBLP | FedAvg | 51.23 | 49.78 | 54.12 | **78.74** |
| | DBLP | FedProx | 51.87 | 49.45 | 52.87 | **79.25** |
| | DBLP | SCAFFOLD | 55.78 | 53.12 | 54.04 | **79.77** |
| | DBLP | FedDF | 50.62 | 54.29 | 53.18 | **77.58** |
| | DBLP | FedNova | 54.21 | 55.81 | 50.12 | **77.80** |
| Physics | PubMed | FedAvg | 54.88 | 57.11 | 53.92 | **63.84** |
| | PubMed | FedProx | 55.14 | 54.29 | 56.32 | **65.61** |
| | PubMed | SCAFFOLD | 55.98 | 58.72 | 52.34 | **65.73** |
| | PubMed | FedDF | 56.41 | 57.29 | 53.47 | **65.61** |
| | PubMed | FedNova | 55.97 | 56.43 | 50.87 | **58.40** |
| DBLP | PubMed | FedAvg | 53.12 | 53.58 | 52.78 | **69.66** |
| | PubMed | FedProx | 54.58 | 52.98 | 54.92 | **71.39** |
| | PubMed | SCAFFOLD | 55.31 | 53.12 | 57.84 | **73.50** |
| | PubMed | FedDF | 50.34 | 54.62 | 54.11 | **78.56** |
| | Citeseer | FedNova | 52.21 | 53.72 | 55.34 | **71.71** |

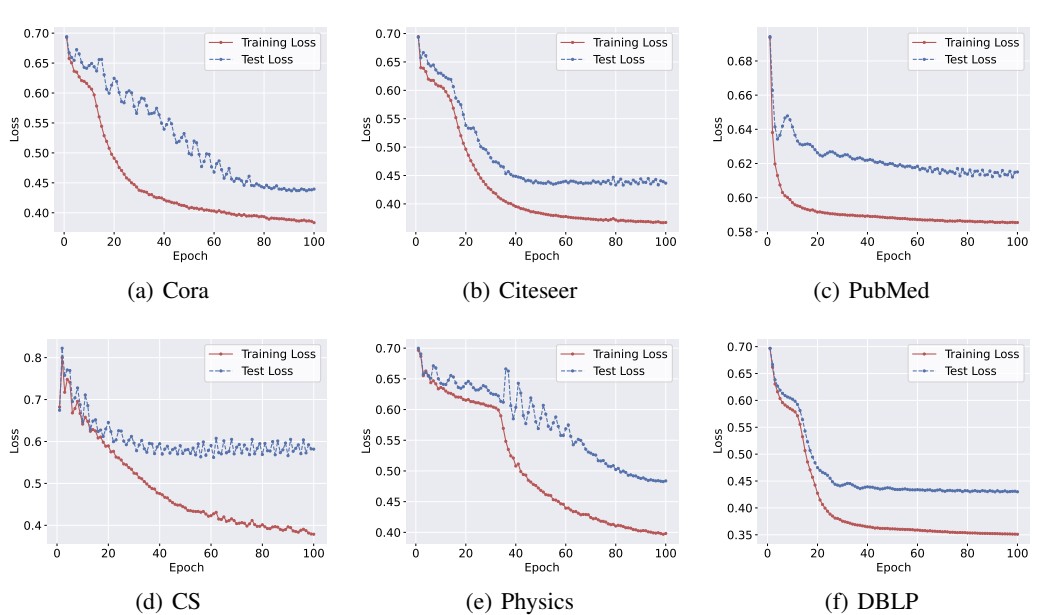

Figure 5: The convergence of `CC-MIA` under membership inference.

curves of the attack model under FedAVG, using the best-matched shadow dataset from Table 1, as illustrated in Fig 5. `CC-MIA` achieves convergence for most datasets within 100 epochs.

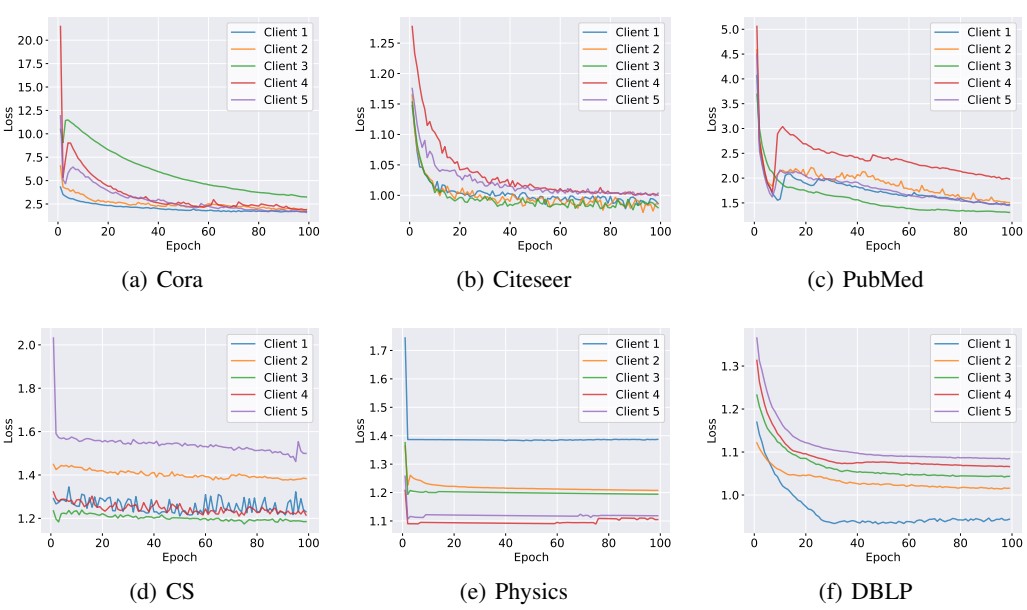

Figure 6: The convergence of `CC-MIA` under client-data identification.

For client-data identification, we examined the convergence of gradient inversion by plotting the loss curves for reconstructing node features and graph structures, as shown in Fig 6. Using a client number of 5 and FedAVG aggregation as an example, we tested gradient inversion with GCN across various datasets. The convergence trend is stable for Cora and PubMed after early oscillation. For CS and Physics, loss sharply declines during the initial epochs before stabilizing in subsequent iterations. In contrast, the DBLP dataset exhibits a gradual convergence, achieving stability after approximately 40

Table 9: Time and space cost of `CC-MIA` on different datasets.

| | Dataset | Cora | Citeseer | PubMed | CS | Physics | DBLP |
|---|---|---|---|---|---|---|---|
| | Train Classifier | 0.07 | 0.08 | 0.21 | 0.20 | 0.37 | 0.19 |
| Time (S) | Grad Inversion | 0.88 | 1.08 | 52.25 | 51.05 | 208.47 | 45.31 |
| | Prototype Match | 1.15 | 1.41 | 8.60 | 8.09 | 14.65 | 7.43 |
| | Train Classifier | 3.02 | 3.02 | 3.02 | 3.02 | 3.02 | 3.02 |
| Space (MB) | Grad Inversion | 6.26 | 9.24 | 317.89 | 268.58 | 951.32 | 275.65 |
| | Prototype Match | 48.09 | 96.45 | 1560.12 | 1793.83 | 5713.85 | 1343.37 |

epochs. Overall, the gradient inversion process for client-membership MIA consistently approximates the original graph, ensuring subsequent prototype construction.

## A.14 COST

We evaluated the time and space overhead of `CC-MIA` across different datasets. Specifically, all experiments were conducted on an Nvidia RTX 3090 Ti GPU, with the number of iterations for both classifier training and gradient inversion set to 100, and the number of clients fixed at 3. The computational overhead was recorded for three components: (1) the attacker's classifier training, which corresponds to the membership inference classification phase in `CC-MIA`, and (2) the gradient inversion and (3) prototype matching, which constitute the two-step process for client-data identification. The results are summarized in Table 9.

Since the parameter size of the classifier is fixed, its space complexity remains constant regardless of the dataset size. However, the time complexity increases with the number of nodes and edge density. As elaborated in Appendix A.4, the runtime of the client-data identification is dominated by the gradient inversion step. Nevertheless, due to the extensive distance computations and storage requirements in the prototype matching step, its space overhead is the largest among the three components.

## A.15 POTENTIAL DEFENSE

Perturbation mechanisms are a common defense against inference attacks. In centralized settings, perturbations are typically applied to GNN model outputs to disrupt adversaries' ability to infer posterior probabilities, hindering MIAs. However, in `CC-MIA`, where attackers leverage gradient information from shadow datasets and target clients via the global model, output perturbations are unusable since clients cannot modify the global model directly.

An alternative defense is to apply perturbations to client node features, disrupting the attacker's ability to exploit gradients derived from these features. While effective, this defense introduces a trade-off, as perturbations degrade the performance of the federated GNN aggregated. Specifically, GCN Kipf & Welling (2017) and FedAvg McMahan et al. (2017) are applied to verify the efficiency of potential defenses.

Differential Privacy (DP) is a well-established perturbation-based mechanism that provides a robust framework for protecting data confidentiality. By introducing carefully calibrated noise during the training or inference, DP significantly reduces the risk of extracting sensitive information from the perturbed data Behnia et al. (2022). Building on this, we define $d_\chi$-Privacy as follows:

**Definition 3** ($d_\chi$-**Privacy**) *Let $X$ denote the input domain, $Y$ the output domain, and $d_\chi$ a distance metric over $X$. A randomized mechanism $M : X \to Y$ satisfies $(\eta d_\chi)$-privacy if, for any two inputs $x, x' \in X$ and any subset $S \subseteq Y$, the following inequality holds:*

$$\frac{\Pr[M(x) \in S]}{\Pr[M(x') \in S]} \leq e^{\eta d_\chi(x,x')}, \tag{29}$$

*where $\eta \geq 0$ represents the privacy budget, balancing the trade-off between privacy and utility.*

In the context of graph data, we define $X$ as the original node features and $X'$ as their corresponding perturbed features, which are introduced to mitigate the effectiveness of `CC-MIA`.

The trade-off of defense is illustrated in Fig 7 and Fig 8.

Applying $d_\chi$-based Chatzikokolakis et al. (2013) perturbations with varying intensities results in a noticeable degradation in the performance of the global GNN. Under the training-set inference scenario, even when the model's performance is severely compromised, CC-MIA remains highly effective in conducting inference attacks. For client-data identification, the attack success rate can only be reduced to near-random levels when the privacy budget is extremely small (*i.e.*, the noise intensity is very high). However, this comes at the unavoidable cost of significant damage to the overall utility and performance of the GNN.

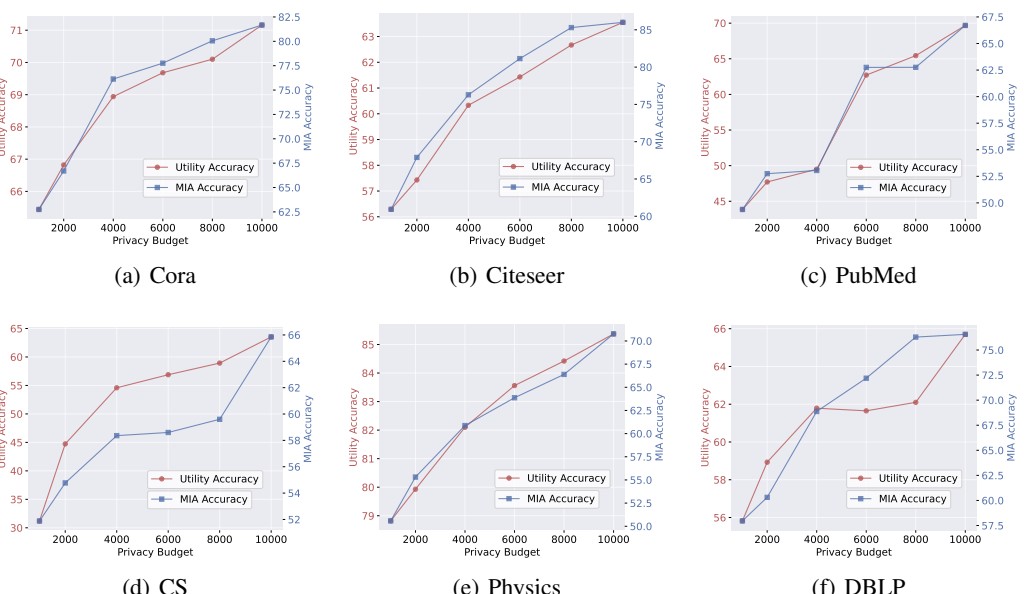

Figure 7: The potential defense against CC-MIA for training-set inference.

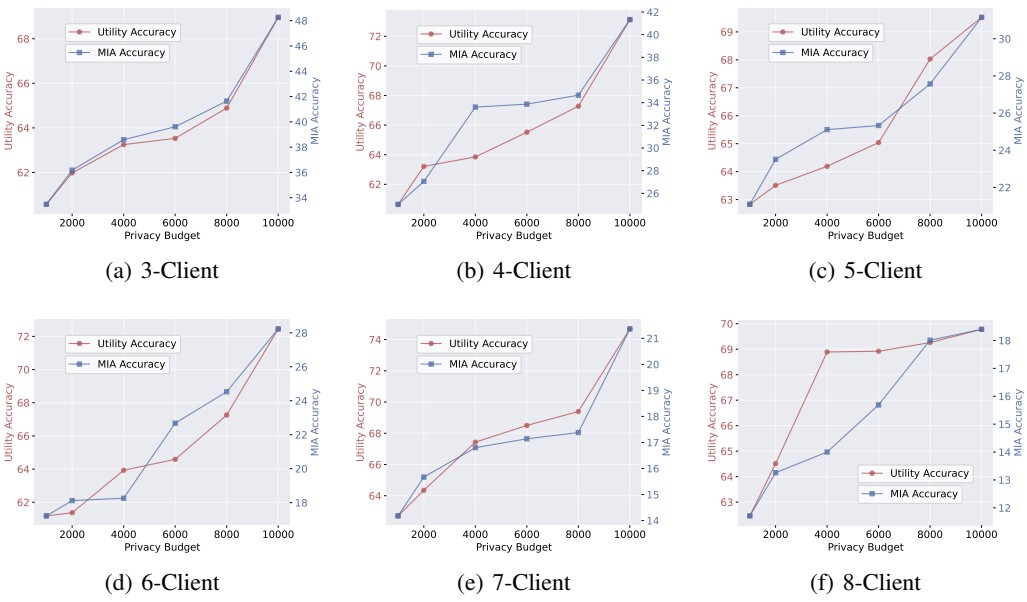

Figure 8: The potential defense against CC-MIA for client-data identification.

Table 10: Sensitivity of gradient inversion to edge density (edges/nodes) under the 5-client setting. "Medium" density approximates the true sparsity for each dataset.

| Dataset | Low (0.5) | | Medium | | High (3.0) | |
|---|---|---|---|---|---|---|
| | AUC↑ | RNMSE↓ | AUC↑ | RNMSE↓ | AUC↑ | RNMSE↓ |
| Cora | 62.31 | 0.0058 | **69.78** | **0.0028** | 64.05 | 0.0049 |
| Citeseer | 65.12 | 0.0042 | **74.23** | **0.0021** | 68.94 | 0.0037 |
| PubMed | 56.83 | 0.0061 | **62.67** | **0.0022** | 58.41 | 0.0053 |
| CS | 68.42 | 0.0008 | **72.71** | **0.0003** | 70.13 | 0.0006 |
| Physics | 59.27 | 0.0009 | **65.43** | **0.0002** | 61.85 | 0.0005 |
| DBLP | 58.64 | 0.0031 | **65.12** | **0.0017** | 60.33 | 0.0028 |

### A.16 DETAILED RELATED WORKS

#### A.16.1 MEMBERSHIP INFERENCE IN FEDERATED LEARNING

Despite its privacy-preserving design, FL remains vulnerable to various inference attacks. Prior studies have shown that adversaries can exploit shared model updates to reconstruct sensitive training data Zhu et al. (2019), infer statistical properties of other clients' datasets Melis et al. (2019), or even synthesize representative inputs Wang et al. (2019). Among these, membership inference attacks (MIAs) pose a fundamental privacy threat by determining whether a specific data sample was used in training. MIAs have significant implications: they can lead to privacy breaches (*e.g.*, revealing a patient's diagnosis by confirming their data was used in a medical model), support compliance auditing (*e.g.*, verifying data deletion under "right to be forgotten" laws), and serve as precursors to more advanced threats such as model extraction. A growing body of work has explored MIAs in the FL setting, beginning with gradient-based attacks that leverage leakage from updates, hidden layer activations, and loss signals Nasr et al. (2019). Recently, new MIAs have focused on methods of stealing hyperparameters Li et al. (2022), shadow training Zhang et al. (2022a), learning logits distribution Yan et al. (2022), and feature construction Liu et al. (2023), *etc*. These attacks have since been extended to various domains, including classification, regression, and recommendation, using both shared model parameters and trends in model outputs over training rounds.

#### A.16.2 MEMBERSHIP INFERENCE IN GRAPH NEURAL NETWORKS

Membership inference attacks (MIAs) have recently been extended to graph neural networks (GNNs), with early efforts adapting the shadow training framework from traditional domains to node classification tasks Olatunji et al. (2021b). One line of work introduced 0-hop, 2-hop, and combined attacks that exploit both node and neighbor posterior distributions to improve inference accuracy He et al. (2021b). Label-only attacks have also been explored, relying solely on predicted labels without access to confidence scores Conti et al. (2022). Beyond node-level inference, graph-level MIAs have been proposed using training-based and threshold-based methods to determine whether an entire graph instance was used during training Wu et al. (2021). Further studies have examined MIAs under adversarial training settings Liu et al. (2022), as well as subgraph-level attacks that construct discriminative features to distinguish target subgraphs Zhang et al. (2022b).

### A.17 REBUTTAL SECTION

#### A.17.1 PARAMETER SENSITIVITY FOR GRADIENT INVERSION

We analyze the impact of edge density on gradient inversion quality. As shown in Table 10, we consider three settings: Low (0.5×), Medium (1.0×, i.e., matching the dataset's natural edge density), and High (3.0×). Results show that inversion performance is best when the reconstructed graph uses the natural edge density—deviations in either direction degrade AUC and increase RNMSE, highlighting the importance of structural realism in FedGNN inversion.

Table 11: True Positive Rate (TPR) of Client-data Identification.

| Dataset | 3-Clts | 4-Clts | 5-Clts | 6-Clts | 7-Clts | 8-Clts | 9-Clts | 10-Clts |
|---------|--------|--------|--------|--------|--------|--------|--------|---------|
| Cora | 0.5484 | 0.3598 | 0.3301 | 0.2959 | 0.2594 | 0.2561 | 0.2113 | 0.1591 |
| Citeseer | 0.5745 | 0.4244 | 0.4092 | 0.3641 | 0.3535 | 0.2491 | 0.2202 | 0.1807 |
| Pubmed | 0.4193 | 0.3813 | 0.2392 | 0.2118 | 0.1602 | 0.1529 | 0.1363 | 0.1265 |
| CS | 0.6993 | 0.4837 | 0.4083 | 0.3593 | 0.2945 | 0.2531 | 0.2004 | 0.1946 |
| Physics | 0.6445 | 0.5551 | 0.3123 | 0.3105 | 0.2764 | 0.2509 | 0.2145 | 0.1717 |
| DBLP | 0.6357 | 0.3899 | 0.3845 | 0.3383 | 0.2943 | 0.2825 | 0.2732 | 0.2324 |

### A.17.2 TPR OF CLIENT-DATA IDENTIFICATION

We also report the True Positive Rate (TPR) for client-data identification in the table above. TPR better reflects the robustness of the attack, as it measures the fraction of correctly identified client-owned samples among all actual positives. Across all datasets and client settings, our TPR consistently and significantly exceeds random guessing, demonstrating the strong and robust performance of CC-MIA even as the number of clients increases.

