# OpenReview forum: "Who Owns This Sample: Cross-Client Membership Inference Attack in Federated Graph Neural Networks"
_ICLR.cc/2026/Conference — Submitted to ICLR 2026_

### Official Review · Reviewer_JtRA · 2025-10-17

**Soundness:** 2
**Presentation:** 2
**Contribution:** 2
**Rating:** 2
**Confidence:** 4

**Summary:**

This paper introduces CC-MIA, a framework for cross-client membership inference attacks (MIAs) in federated graph neural networks (FedGNNs). It reformulates traditional MIAs to not only determine if a node was part of the training set but also attribute it to the specific client that owns it. The attack leverages a threat model where an adversarial client observes global model updates and eavesdrops on other clients' gradients. Using publicly available shadow datasets from the same domain, CC-MIA combines shadow-based training for membership inference, gradient inversion to reconstruct client subgraphs, and prototype-based matching for client attribution. Experiments are conducted across six benchmark datasets and five federated learning schemes, demonstrating the effectiveness of the proposed method.

**Strengths:**

+ The research question is reasonable, considering that the client-wise ownership is important in FL MIA.
+ Various experimental evaluation claimed on multiple datasets (six benchmarks) and federated schemes (five), with quantitative improvements over baselines.

**Weaknesses:**

- The threat model is too strong, especially for gradient eavesdropping. It may not be hold in some FL settings, e.g., secure aggregation, cross-device FL (the number client is large). Besides, I can not get how the malicious client could possibly another client's gradients in practice.
- The assumption of shadow dataset requires more evaluations. It is important in FL, considering the client-side datasets are normally private, and it may be hard to find the in-domain public data.
- It would be better for the authors to provide more details on the related works. There have been many works on MIA in GNN, the authors should make a detailed comparison to present the novelty of designs. Besides, the authors should highlight the differences between centralized and FL MIA.
- Assuming the clients share gradients instead of weight updates is limited. For practical FL, the clients do not directly share gradients, they normally share weight update after multiple rounds of mini-batch local updates. This is also important for the performance gradient inversion attacks.
- The authors should explicitly make evaluations on non-iid settings, and it is important to FL and MIA (e.g., transferability).
- The performance of client-wise ownership inference is not that effective. Though it is higher that the random-guess, there is (possibly) still a very high TPR for the attacker, which will comprise the confidence.

**Questions:**

1. Why the assumption on gradient eavesdropping is reasonable?
2. More evaluations on shadow dataset and non-iid settings are needed.
3. The novelty and differences between the proposed MIA and previous MIAs on GNN.
4. Is the attack only effective against raw gradients instead of multiple rounds of mini-batch local updates?
5. Report the TPR of client-wise ownership inference.

**Details Of Ethics Concerns:**

None.

---

> ### Author Response · Authors · 2025-11-20
>
> #### Thanks to the reviewer ``JtRA``'s valuable comments! We will respond to address all the weaknesses and questions.
> > #### [**Weakness 1**]: The threat model is too strong, especially for gradient eavesdropping. It may not be hold in some FL settings, e.g., secure aggregation, cross-device FL (the number client is large). Besides, I can not get how the malicious client could possibly another client's gradients in practice.
> > #### [**Question 1**]: Why the assumption on gradient eavesdropping is reasonable?
> > #### [**Response**]: Your concern regarding the threat model is very important. We clarify that our work assumes the attacker is a malicious client operating within a standard federated learning architecture without Secure Aggregation, who can eavesdrop on the plaintext gradients uploaded by other clients to the server.  This assumption is widely adopted in privacy attack literature, such as in DLG and iDLG, and does not constitute an overly strong threat model.
> > #### In typical cross-silo settings (e.g., collaborations among hospitals or enterprises), the number of participants is limited, communication links are often unencrypted, and mutual trust is incomplete.  Under these conditions, gradient eavesdropping (e.g., via network sniffing) becomes a low-cost and highly feasible passive attack vector.  Recent works [1–4] also commonly treat this as a realistic threat.
> > #### Importantly, our goal is **not to break cryptographic defenses** such as Secure Aggregation, but rather to expose the genuine privacy risks present in current FedGNN deployments that, for reasons of performance or compatibility, often omit encrypted aggregation.  Indeed, once Secure Aggregation is enabled, raw gradients are hidden from all parties, and CC-MIA immediately becomes ineffective, which precisely demonstrates the efficacy of such defenses and delineates the valid scope of our threat model.
> > #### Therefore, under practical federated graph learning scenarios without encryption, the gradient eavesdropping assumption is reasonable, common, and fully aligned with the mainstream paradigm of privacy attacks.
> > #### [1] Wu H, Fang Y, Li N, et al. Secret Key Generation With Untrusted Internal Eavesdropper: Token-Based Anti-Eavesdropping[J]. IEEE Transactions on Information Forensics and Security, 2025.
> > #### [2] Hu R, Gong Y, Guo Y. Federated Learning with Sparsification-Amplified Privacy and Adaptive Optimization[C]//Proceedings of the Thirtieth International Joint Conference on Artificial Intelligence. 2021.
> > #### [3] Xu C, Neglia G. What else is leaked when eavesdropping federated learning?[C]//CCS workshop Privacy Preserving Machine Learning (PPML). 2021.
> > #### [4] Meindl R, Moser B A. Measuring overhead costs of federated learning systems by eavesdropping[C]//International Conference on Database and Expert Systems Applications. Cham: Springer Nature Switzerland, 2023: 33-42.
>
> > #### [**Weakness 2**]: The assumption of shadow dataset requires more evaluations. It is important in FL, considering the client-side datasets are normally private, and it may be hard to find the in-domain public data.
> > #### [**Question 2**]: More evaluations on shadow dataset and non-iid settings are needed.
> > #### [**Response**]: In many real-world FedGNN applications, such as academic citation networks, social platforms, and recommendation systems, publicly available graph datasets with similar semantic structures (e.g., Cora, DBLP, Reddit) are commonly accessible. This provides a practical foundation for constructing effective shadow models.  Even in extreme cases, it is reasonable to assume that a malicious client participating in federated aggregation has knowledge of the target data's category domain (e.g., "research topics" or "user interests").  Under this assumption, the attacker can build a substitute dataset from the same domain, and the attack may still succeed.
> > #### **This is empirically validated in our cross-dataset attack experiments (see Table 5): even when feature spaces are not perfectly aligned, CC-MIA achieves strong membership inference performance as long as the class semantics and graph topology exhibit sufficient similarity**.

---

> ### Author Response · Authors · 2025-11-20
>
> > #### [**Weakness 3**]: It would be better for the authors to provide more details on the related works. There have been many works on MIA in GNN, the authors should make a detailed comparison to present the novelty of designs. Besides, the authors should highlight the differences between centralized and FL MIA.
> > #### [**Question 3**]: The novelty and differences between the proposed MIA and previous MIAs on GNN.
> > #### [**Response**]: Our proposed CC-MIA fundamentally differs from traditional centralized MIAs in **threat model, information utilization, and privacy risk granularity**:
> | Dimension | Traditional Centralized MIA | CC-MIA (Federated GNN Setting) |
> |----------|------------------------------|--------------------------------|
> | **Attacker** | External adversary or server with full model access | Malicious client participating in training (not the server) |
> | **Accessible Information** | Can observe the target model’s output for individual samples (e.g., logits or softmax probabilities) | Only has access to aggregated gradients uploaded by clients; no raw data, graph structure, or per-sample predictions |
> | **Attack Input** | Original sample paired with model response | Virtual graph $(\hat{X}, \hat{A})$ reconstructed via gradient inversion |
> | **Inference Granularity** | Sample-level: determines whether a specific data point was in the training set | Class-client joint level: determines (1) whether a sample belongs to the training distribution and (2) whether nodes of a given class appear in a specific client’s local subgraph |
> | **Key Assumptions** | Requires an i.i.d. shadow dataset aligned with the target data distribution | Leverages cross-client collaborative signals and prototype separability inherent in GNNs under non-IID graph partitions |
>
> > #### [**Weakness 4**]: Assuming the clients share gradients instead of weight updates is limited. For practical FL, the clients do not directly share gradients, they normally share weight update after multiple rounds of mini-batch local updates. This is also important for the performance gradient inversion attacks.
> > #### [**Question 4**]: Is the attack only effective against raw gradients instead of multiple rounds of mini-batch local updates?
> > #### [**Response**]: Thank you for this insightful question.  We emphasize that our attack does not assume access to raw, per-step gradients, it's a common misconception, but instead operates on the aggregated weight updates ($\Delta W$) that clients upload after performing multiple local mini-batch epochs (e.g., $E = 5$ local epochs per round), exactly as in standard FedAvg.
> > #### The attack is launched only once, at a late communication round (e.g., round 180 out of $T = 200$), when the global FedGNN model has nearly converged.  At this stage, the client’s uploaded update $\Delta W$ reflects the cumulative effect of many local optimization steps and encodes strong, class-discriminative semantic signals due to the emergence of well-separated prototype structures in the GNN embedding space.
> > #### Thus, the attack targets the "aggregated signal after multiple local updates," not raw single-step gradients, which fully aligns with real-world federated learning workflows.
>
> > #### [**Weakness 5**]: The authors should explicitly make evaluations on non-iid settings, and it is important to FL and MIA (e.g., transferability).
> > #### [**Response**]: Thank you for this important point. In fact, all of our experiments were conducted in a real-world non-IID setting, where each client held a subgraph containing only a subset of the class, which precisely because of homomorphic induced clustering in real-world graphs, as described in Section 5.2.
> > #### This natural non-IID setting is not a limitation but a **key driving factor for CC-MIA**: because the clients focus on different semantic regions, their aggregated gradients retain strong class-conditional signals, making prototype-based reasoning feasible. Our experimental results all reflect the performance under this non-IID condition.
> > #### In contrast, the challenges in heterogeneous graphs (for example, multiple node/edge types) are non-IID orthogonal and related to the complexity of the graph schema; We will discuss this as the future direction. However, for the standard homogeneous graph in the real non-IID (the main FL scenario), our assessment is comprehensive and consistent with FL practice.

---

> ### Author Response · Authors · 2025-11-20
>
> > #### [**Weakness 6**]: The performance of client-wise ownership inference is not that effective. Though it is higher that the random-guess, there is (possibly) still a very high TPR for the attacker, which will comprise the confidence.
> > #### [**Question 5**]: Report the TPR of client-wise ownership inference.
> > #### [**Response**]: Thank you for your attention. Indeed, the True Positive Rate (TPR) is a more informative metric than Accuracy for evaluating attack robustness. We report the TPR of client-wise ownership inference, which remains significantly higher than random guessing. This clearly demonstrates the robustness of our CC-MIA. We will update the following table in A.17 Rebuttal Section of the paper accordingly.
> | TPR | 3-Clts | 4-Clts | 5-Clts | 6-Clts | 7-Clts | 8-Clts | 9-Clts | 10-Clts |
> |-----|--------|--------|--------|--------|--------|--------|--------|---------|
> | Cora | 0.5484 | 0.3598 | 0.3301 | 0.2959 | 0.2594 | 0.2561 | 0.2113 | 0.1591 |
> | Citeseer | 0.5745 | 0.4244 | 0.4092 | 0.3641 | 0.3535 | 0.2491 | 0.2202 | 0.1807 |
> | Pubmed | 0.4193 | 0.3813 | 0.2392 | 0.2118 | 0.1602 | 0.1529 | 0.1363 | 0.1265 |
> | CS | 0.6993 | 0.4837 | 0.4083 | 0.3593 | 0.2945 | 0.2531 | 0.2004 | 0.1946 |
> | Physics | 0.6445 | 0.5551 | 0.3123 | 0.3105 | 0.2764 | 0.2509 | 0.2145 | 0.1717 |
> | DBLP | 0.6357 | 0.3899 | 0.3845 | 0.3383 | 0.2943 | 0.2825 | 0.2732 | 0.2324 |
>
> **If you find our clarifications valuable, we would be grateful for your positive consideration during the evaluation process!**

---

> > ### Comment · Reviewer_JtRA · 2025-11-28
> >
> > Thank you for your response, and most of my concerns have been addressed. However, I still find the assumption of malicious clients to be rather strong. Unlike malicious servers (DLG), I struggle to imagine how clients could effectively hijack another client's gradients in practical scenarios. It would be better if the authors could provide more explanations.

---

> > > ### Author Response · Authors · 2025-11-28
> > >
> > > Thank you for your valuable feedback.
> > >
> > > > #### We would like to clarify that eavesdropping is a common attack in the communication process [2-6]. The essence of federated learning (FL) lies in the exchange and aggregation of information between clients and the server through communication.  In most practical FL implementations, clients transmit gradient-based updates, such as model deltas to the central server [1].
> > > > #### Since FL inherently relies on communication, it is inevitably exposed to eavesdropping risks. Indeed, eavesdropping through communication channels is a common and well-recognized threat model. The eavesdropping attack model can be formalized with the following key equations:
> > > > #### Eavesdropper's Estimation Dynamics:
> > > \begin{equation}
> > > \hat{w}^{k} _{i, {adv}} = \hat{w}^{k-1} _{i, {adv}} + \alpha^{k} _{i} q^{k-1} _{i}
> > > \end{equation}
> > > > #### where $\hat{w}^{k} _{i,\text{adv}}$ denotes the adversary's estimate of device $i$'s weight at iteration $k$; $q^{k-1} _{i}$ denotes the difference weight actually transmitted by device $i$; $\alpha^{k} _{i}$ denotes a Bernoulli random variable indicating whether the transmission is intercepted.
> > > > #### Interception Success Probability:
> > > \begin{equation}
> > > \Pr(\alpha^{k}_{i} = 1) = \bar{\alpha}_i \in (0,1)
> > > \end{equation}
> > > where $\bar{\alpha}_i$ dednotes the probability that the adversary successfully eavesdrops on device $i$'s communication.
> > > > #### Prior work has extensively studied eavesdropping attacks across different devices in distributed settings. FL itself was originally deployed by Google on smartphones for privacy-preserving on-device learning [7, 8]. However, many existing studies overlook the communication-centric nature of FL and consequently underestimate the threat posed by eavesdropping. In our work, we highlight that eavesdropping can severely compromise FL systems, and we bring this critical security concern to the forefront.  Espeically many GNN applications running on the mobile device for supporting the recommendation systems, which is also the important senario for FL. Therefore, evasdropping presents the realistic threats for FedGNN.
> > > > #### [1] Xu Lei, Xu Danya, Yi Xinlei, Deng Chao, Chai Tianyou, Yang Tao. Decentralized Federated Learning Algorithm Under Adversary Eavesdropping[J]. IEEE/CAA Journal of Automatica Sinica, 2025, 12(2): 448-456. SCI IF: 19.2
> > > > #### [2] Lu Li, Chen Meng, Yu Jiadi, Ba Zhongjie, Lin Feng, Han Jinsong, Zhu Yanmin, Ren Kui. An imperceptible eavesdropping attack on WiFi sensing systems[J]. IEEE/ACM Transactions on Networking, 2024, 32(5): 4009-4024. CCF-A
> > > > #### [3] Wang Chao, Lin Feng, Liu Tiantian, Zheng Kaidi, Wang Zhibo, Li Zhengxiong, Huang Min-Chun, Xu Wenyao, Ren Kui. mmEve: eavesdropping on smartphone's earpiece via COTS mmWave device[C]//Proceedings of the 28th Annual International Conference on Mobile Computing And Networking. 2022: 338-351. CCF-A
> > > > #### [4] Huang Peng, Wei Yao, Cheng Peng, Ba Zhongjie, Lu li, Lin Feng, Wang Yang, Ren Kui. Phoneme-Based Proactive Anti-Eavesdropping With Controlled Recording Privilege[J]. IEEE Transactions on Dependable and Secure Computing, 2024. CCF-A
> > > > #### [5] Ba Zhongjie, Zheng Tianhang, Qin Zhan, Yu Hanlin, Liu Liu, Li Baochun, Liu Xue, Ren Kui. Accelerometer-based smartphone eavesdropping[C]//Proceedings of the 26th Annual International Conference on Mobile Computing and Networking. 2020: 1-2. CCF-A
> > > > #### [6] Wang Lei, Chen Meng, Lu Li, Ba Zhongjie, Lin Feng, Ren kui. Voicelistener: A training-free and universal eavesdropping attack on built-in speakers of mobile devices[J]. Proceedings of the ACM on Interactive, Mobile, Wearable and Ubiquitous Technologies, 2023, 7(1): 1-22. CCF-A
> > > > #### [7] Yang Qiang, Liu Yang, Chen Tianjian, Tong Yongxin. Federated machine learning: Concept and applications[J]. ACM Transactions on Intelligent Systems and Technology (TIST), 2019, 10(2): 1-19.
> > > > #### [8] Yang Wenti, Wang Naiyu, Guan Zzhitao, Wu Longfei, Du Xiaojiang, Guizani Mohsen. A practical cross-device federated learning framework over 5g networks[J]. IEEE Wireless Communications, 2022, 29(6): 128-134.
> > >
> > > We hope the above discussion and key references adequately address your concern.  These clarifications will be incorporated into the revised manuscript.  If you find the issue sufficiently resolved, we would greatly appreciate your consideration in revising your score upward.

---

### Official Review · Reviewer_8CZA · 2025-10-29

**Soundness:** 3
**Presentation:** 2
**Contribution:** 3
**Rating:** 4
**Confidence:** 4

**Summary:**

The paper proposes a method for membership inference attacks in GNNs (Graph Neural Networks) under the Federated Learning (FL) setup. Specifically, it assumes that one client in the FL setup is adversarial, and tries to infer two types of information: first, whether a given node was part of the training graph (membership inference), and second, which client owns the node (cross-client attribution). To solve for these, the paper proposes a framework called CC-MIA (Cross-Client Membership Inference Attack). While membership inference attacks have been shown for GNNs and for FL in the past, the setup explored in this paper is novel.

**Strengths:**

1. The paper explores a novel attack vector; membership attacks on GNNs in FL is a previously unexplored area.
2. The attack model used in the paper is based on commonly accepted assumptions used in the FL and MIA literature, such as access to a shadow dataset, gradient eavesdropping capabilities, etc.
3. The paper includes comprehensive experimentation including different types of GNNs, FL algorithms, and graph datasets.

**Weaknesses:**

1. The first part of the attack, which is membership inference, is conceptually not much different from a membership inference attack on a centralized GNN, such as Olatunji et al. [a]. The paper could benefit from explicitly stating the challenges that FL brings into the picture.
2. The second part of the attack uses a pseudo-graph (as mentioned in line 256), based on which the optimization in eq(12) is carried out. However, it is unclear how this pseudograph is obtained or generated.
3. At several points in the paper, the underlying setup and assumptions have not been clearly stated, rendering it difficult to understand the work. For instance,
    - In Section 3.1, the Federated GNN setup could be explained better: Is this a horizontal FL setup where all clients have unique nodes? Do clients' subgraphs have connecting edges?
    - The objective function in eq(12) does not mention the optimization variable.
    - Certain variables have not been defined, for e.g., $D$ in line 217, $c$ in line 226, $x_t^a$ and $x_t^c$ in line 240.
    - Value of hyperparamter $\gamma$ in line 236 has not been stated.
    - There is an overuse of notation at several places, e.g. $x_t^a$ in line 240, $D$ as diagonal matrix in line 266, $D$ in line 217,

[a] Iyiola E Olatunji, Wolfgang Nejdl, and Megha Khosla. Membership inference attack on graph neural networks. In 2021 Third IEEE International Conference on Trust, Privacy and Security in Intelligent Systems and Applications (TPS-ISA), pp. 11–20. IEEE, 2021a.

**Questions:**

1. How does the proposed membership inference attack differ from that on a centralized GNN, in terms of unique challenges encountered and addressed?
2. What are the assumptions for the shadow dataset? Does it also have subgraphs like the original training dataset? Also, is it possible for the adversarial client to train the shadow network on its own subgraph dataset, instead of the shadow dataset?
3. How frequently (for how many FL rounds) is the gradient inversion step performed?
4. How is the proxy term $\zeta_t$ estimated?
5. How does the value of the hyperparameter $\gamma$ affect gradient inversion?

---

> ### Author Response · Authors · 2025-11-20
>
> #### Thanks to the reviewer ``8CZA``'s valuable comments! We will respond to you to address all the weaknesses and questions.
> > #### [**Weakness 1**]: The first part of the attack, which is membership inference, is conceptually not much different from a membership inference attack on a centralized GNN, such as Olatunji et al. [a]. The paper could benefit from explicitly stating the challenges that FL brings into the picture.
> > #### [a] Iyiola E Olatunji, Wolfgang Nejdl, and Megha Khosla. Membership inference attack on graph neural networks. In 2021 Third IEEE International Conference on Trust, Privacy and Security in Intelligent Systems and Applications (TPS-ISA), pp. 11–20. IEEE, 2021a.
> > #### [**Question 1**]: How does the proposed membership inference attack differ from that on a centralized GNN, in terms of unique challenges encountered and addressed?
> > #### [**Response**]: Our proposed CC-MIA fundamentally differs from traditional centralized MIAs in **threat model, information utilization, and privacy risk granularity**:
> | Dimension | Traditional Centralized MIA | CC-MIA (Federated GNN Setting) |
> |----------|------------------------------|--------------------------------|
> | **Attacker** | External adversary or server with full model access | Malicious client participating in training (not the server) |
> | **Accessible Information** | Can observe the target model’s output for individual samples (e.g., logits or softmax probabilities) | Only has access to aggregated gradients uploaded by clients; no raw data, graph structure, or per-sample predictions |
> | **Attack Input** | Original sample paired with model response | Virtual graph $(\hat{X}, \hat{A})$ reconstructed via gradient inversion |
> | **Inference Granularity** | Sample-level: determines whether a specific data point was in the training set | Class-client joint level: determines (1) whether a sample belongs to the training distribution and (2) whether nodes of a given class appear in a specific client’s local subgraph |
> | **Key Assumptions** | Requires an i.i.d. shadow dataset aligned with the target data distribution | Leverages cross-client collaborative signals and prototype separability inherent in GNNs under non-IID graph partitions |
>
> > #### [**Weakness 2**]: The second part of the attack uses a pseudo-graph (as mentioned in line 256), based on which the optimization in eq(12) is carried out. However, it is unclear how this pseudograph is obtained or generated.
> > #### [**Response**]: The pseudo-graph (i.e., $\hat{X}, \hat{A}$) mentioned on line 256 is not pre-defined but iteratively reconstructed from gradients through an optimization process. Specifically, starting from randomly initialized $\hat{X}^{(0)}$ and $\hat{A}^{(0)}$, we solve the optimization problem in Equation 12 of the paper to progressively refine the virtual graph until its generated gradients closely match the observed client gradient $G_k$.
>
> > #### [**Weakness 3**]: At several points in the paper, the underlying setup and assumptions have not been clearly stated, rendering it difficult to understand the work. For instance,
> > #### In Section 3.1, the Federated GNN setup could be explained better: Is this a horizontal FL setup where all clients have unique nodes? Do clients' subgraphs have connecting edges?
> > #### The objective function in eq(12) does not mention the optimization variable.
> > #### Certain variables have not been defined, for e.g., D in line 217, c in line 226, $x_t^a$ and $x_t^c$ in line 240.
> > #### Value of hyperparamter $\gamma$ in line 236 has not been stated.
> > #### There is an overuse of notation at several places, e.g. $x_t^a$ in line 240, D as diagonal matrix in line 266, in line 217,
> > #### [**Response**]:
> > - #### This work adopts a **horizontal federated learning** setting, where each client holds a disjoint subset of nodes, and these **subgraphs are not connected by any inter-client edges**. We have added explanations in Section 3.1.
> > - #### The optimization variables in Equation 12 are $\hat{X}$ and $\hat{A}$, and the objective is achieved by minimizing the reconstructed loss $\hat{\mathbb{L}}$. We have added explanations in the paper.
> > - #### D in line 217 denotes the global training dataset. $c$ in line 226 denotes the node class. $x_t^c$ typically denotes the true client update at round $t$ and $x_t^a$ denotes the adversary’s estimate of that update at round $t$. We have added explanations in the paper.
> > - #### Note that the hyperparameter $\gamma$ (line 236) can range from 0.3 to 0.6; in our experiments, we set it to 0.5. Importantly, the eavesdropping process and the attack execution are decoupled, so the choice of $\gamma$ does not affect attack performance.
> > - #### We have replaced the previously reused symbol "estimate" on line 240 with $z$ to avoid ambiguity and replace $D$ to $\mathbb{D}$ in line 217.

---

> ### Author Response · Authors · 2025-11-20
>
> > #### [**Question 2**]: What are the assumptions for the shadow dataset? Does it also have subgraphs like the original training dataset? Also, is it possible for the adversarial client to train the shadow network on its own subgraph dataset, instead of the shadow dataset?
> > #### [**Response**]: (1) The core assumption of our shadow dataset is that **it shares semantic structure and class distribution similarity with the target training data** (e.g., both being citation networks) without requiring identical feature spaces or exact graph topology. Specifically, the shadow dataset can be either a full public graph (e.g., DBLP or Physics) or a collection of subgraphs partitioned in the same federated manner as the target clients. In our experiments, we compare these two settings (see Table 2):  **CC-MIA**: Uses a complete public graph (e.g., DBLP) as the shadow dataset, achieving the best attack performance. **CC-MIA (subgraph shadow)**: Trains the shadow model on subgraphs generated via the same client-level partitioning; this leads to noticeably degraded performance.
> > #### (2)While it is theoretically possible for an adversarial client to train a shadow model using only its own local subgraph, this approach is practically limited. In typical FedGNN scenarios, each client holds **non-IID** data, for instance, one client may contain only "machine learning" papers and lack any "database" papers. If the attacker (a malicious client) trains its shadow model solely on its own subgraph, the model cannot learn gradient behaviors for unseen classes. However, membership inference in our setting requires determining whether any class appears in any client’s local data. If the shadow model has never encountered a particular class during training, it cannot construct a meaningful prototype for that class, leading to failed inference for those missing categories.
>
> > #### [**Question 3**]: How frequently (for how many FL rounds) is the gradient inversion step performed?
> > #### [**Response**]: The gradient inversion step was run for 100 epochs. On an NVIDIA RTX 4090, the attack on Cora achieved a speed of 185.46 epochs per second.
>
> > #### [**Question 4**]: How is the proxy term $\zeta_t$ estimated?
> > #### [**Response**]: The proxy term $\zeta_t$ serves as a surrogate signal to approximate the client's state change in rounds where the attacker fails to eavesdrop on the client update (i.e., when $\mu_t = 0$). This estimation is based on a lightweight local proxy rule, and the procedure is as follows:
> > #### 1. State representation:
> > #### The attacker maintains an estimate $z_t^a$ of the client’s local embedding state, where the superscript $a$ denotes the adversary’s estimate and $c$ denotes the client’s true state $z_t^c$.
> > #### 2. When $\mu_t = 1$ (eavesdropping succeeds):
> > #### The attacker directly uses the observed update $\xi_t$—derived from gradient inversion (e.g., features or prototypes of the reconstructed virtual graph)—to refine its estimate:
> > #### $z_t^a = z_{t-1}^a + \xi_t$.
> > #### 3. When $\mu_t = 0$ (eavesdropping fails):
> > #### Since $\xi_t$ is unavailable, the attacker substitutes it with the proxy term $\zeta_t$:
> > #### $z_t^a = z_{t-1}^a + \zeta_t$.
> > #### 4. Construction of $\zeta_t$ (strategy used in our experiments):
> > #### We adopt a simple yet effective zero-order hold strategy:
> > #### $\zeta_t = 0$,
> > #### which assumes that the client’s state remains unchanged during unobserved rounds. This approach has proven sufficiently effective in typical FedGNN settings, where client updates are infrequent and data distributions evolve relatively slowly.

---

> ### Author Response · Authors · 2025-11-20
>
> > #### [**Question 5**]: How does the value of the hyperparameter $\gamma$ affect gradient inversion?
> > #### [**Response**]: The hyperparameter $\gamma$ primarily affects the eavesdropping process rather than the gradient inversion itself.  Since $\gamma$ controls the eavesdropping mechanism and the attack execution is decoupled from the eavesdropping process, the choice of $\gamma$ does not directly impact the gradient inversion performance. The gradient inversion step operates independently on the collected gradient information, regardless of how $\gamma$ influences the eavesdropping success rate.
> > #### The impact on gradient inversion is fundamentally mediated through controlling the **edge density** of the reconstructed graph. In practice, it suffices to predefine a reasonable edge density range based on the type of target graph (e.g., citation or social networks). To this end, we have added an edge sparsity sensitivity experiment. The results show that gradient inversion performs best when the reconstructed graph's edge density closely matches that of the original graph. We have also synchronized this experiment to the A.17 Rebuttal SECTION of the paper.
> | Dataset | Low (0.5×) AUC↑ | Low RNMSE↓ | Medium AUC↑ | Medium RNMSE↓ | High (3.0×) AUC↑ | High RNMSE↓ |
> |---------|------------------|------------|--------------|----------------|-------------------|-------------|
> | Cora    | 62.31            | 0.0058     | 69.78        | 0.0028         | 64.05             | 0.0049      |
> | Citeseer| 65.12            | 0.0042     | 74.23        | 0.0021         | 68.94             | 0.0037      |
> | PubMed  | 56.83            | 0.0061     | 62.67        | 0.0022         | 58.41             | 0.0053      |
> | CS      | 68.42            | 0.0008     | 72.71        | 0.0003         | 70.13             | 0.0006      |
> | Physics | 59.27            | 0.0009     | 65.43        | 0.0002         | 61.85             | 0.0005      |
> | DBLP    | 58.64            | 0.0031     | 65.12        | 0.0017         | 60.33             | 0.0028      |
>
> **If you find our clarifications helpful, we would greatly appreciate your favorable consideration in the evaluation process.**

---

> ### Author Response · Authors · 2025-11-28
>
> We sincerely thank the reviewer ``8CZA`` for your time and valuable feedback on our work! We would be deeply grateful if you could kindly review our response.

---

> ### Comment · Reviewer_8CZA · 2025-11-28
>
> I would like to thank the authors for their effort to address my concerns and comments. I am now convinced about the difference wrt centralized MIA on GNNs, and my doubts related to pseudo-graph generation and shadow dataset have also been clarified. I have accordingly raised my score to 6.

---

> > ### Author Response · Authors · 2025-11-28
> >
> > We sincerely thank you once again for your valuable time and the insightful suggestions on our work!

---

### Official Review · Reviewer_NhdL · 2025-11-01

**Soundness:** 3
**Presentation:** 3
**Contribution:** 3
**Rating:** 6
**Confidence:** 2

**Summary:**

This paper proposes a membership inference attack method for federated graph learning. Specifically, the proposed method successfully achieves two types of membership inference: training set membership inference and client-level membership inference. Technically, the authors leverage gradient inversion technology to implement the attack. Extensive experiments conducted in this work demonstrate the effectiveness of the proposed method.

**Strengths:**

1. The paper pioneers the proposal of a membership inference attack method tailored for federated graph learning, filling the research gap in this specific domain.
2. For the client-data identification problem, the authors design a gradient inversion reconstruction method exclusively for graph data. This method effectively reconstructs the original graph structure from gradient information, showing strong targeted performance for graph-specific characteristics.
3. Compared with the contrastive experiments conducted under the centralized setting, the attack method proposed in this paper demonstrates a significant advantage in success rate, fully verifying its superiority in federated scenarios.

**Weaknesses:**

1. The method for the membership inference attack part lacks special design targeted at graph data which may limit the method’s adaptability and effectiveness in graph-specific scenarios.
2. The paper does not clearly explain on the differences between the method used in the membership inference attack part and the methods applied in the centralized setting. Key distinctions in aspects such as data access constraints, gradient utilization patterns, and attack optimization objectives remain unaddressed, making it difficult for readers to understand the method’s novelty in federated scenarios.
3. The client-data identification part utilizes virtual graph data, but the generation process of this virtual graph is not explained in detail.

**Questions:**

1. Are the node features of Cora and DBLP (shadow dataset for Cora) consistent? Why can DBLP be used for Cora’s membership inference?
2. Do category prototypes from the shared global GNN have spatial consistency? Is using them for client-level data inference reasonable?
3. What differences exist between the proposed membership inference method and traditional centralized ones? What unique advantages does it have?

---

> ### Author Response · Authors · 2025-11-20
>
> #### Thanks to the reviewer ``NhdL``'s valuable comments! We will write our response to address all the weaknesses and questions.
> > #### [**Weakness 1**]: The method for the membership inference attack part lacks special design targeted at graph data which may limit the method’s adaptability and effectiveness in graph-specific scenarios.
> > #### [**Response**]: Our membership inference attack follows the classic "shadow model + attack model" framework, but its key innovation lies in the fact that **the features and decision logic are entirely derived from graph neural network-specific gradient and embedding structures, rather than directly adapting standard methods from image or text domains**. Our graph-based membership inference incorporates several specialized designs:
> > #### **The attack features originate from gradients of GNN layers**. Specifically, we exploit node-level gradients generated during GNN message passing, such as those from aggregator weights or adjacency-aware embedding updates, which inherently encode topological information. These gradients are fundamentally different from the sample-wise gradients in CNNs or MLPs, where inputs lack explicit relational structure.
> > #### **The prototype separability assumption is rooted in graph representation learning**. The notion of "separable prototypes" stems from our observation that, even under non-IID federated settings, GNNs naturally form distinct class clusters in the embedding space due to their neighborhood aggregation mechanism. This clustering behavior is a unique property of graph representation learning and does not generally hold for conventional models trained on independent and identically distributed data.
> > #### **The collaborative client mechanism leverages graph connectivity**. CC-MIA benefits from multiple clients that share the same class labels, jointly contributing gradient signals. Such collaboration arises naturally in federated graph learning because real-world graphs often exhibit latent cross-client connections, eg, edges between users across different devices in a social network, providing additional signal that enhances attack effectiveness.
> > #### CC-MIA is applicable to the majority of homogeneous graph scenarios commonly studied in federated graph learning. For a few specialized cases, such as heterogeneous graphs, significant challenges arise: multiple node and edge types lead to semantically mixed gradients, the absence of a unified label space complicates class alignment, structural heterogeneity weakens prototype separability, and it becomes difficult to construct shadow datasets that match the true heterogeneous distribution. Together, these factors undermine the core assumptions of "class-consistent gradients" and "separable prototypes." Exploring privacy attacks and defenses tailored to heterogeneous graphs remains an important direction for future work. We will incorporate all these clarifications into the main text.

---

> ### Author Response · Authors · 2025-11-20
>
> > #### [**Weakness 2**]: The paper does not clearly explain on the differences between the method used in the membership inference attack part and the methods applied in the centralized setting. Key distinctions in aspects such as data access constraints, gradient utilization patterns, and attack optimization objectives remain unaddressed, making it difficult for readers to understand the method’s novelty in federated scenarios.
> > #### [**Response**]: Thank you for your careful observation regarding the experimental setup. If the server were an untrusted party, the setting would indeed collapse to a centralized scenario. In our work, however, the attacker is positioned as a malicious client who possesses eavesdropping capability to intercept gradients uploaded by other clients and leverages its own shadow dataset to mount the attack. Specifically, the key distinctions are as follows:
> > #### **Data access constraints**.
> > - #### **Centralized MIA**: The attacker has access to auxiliary data drawn from the same distribution as the target model’s training set and can directly observe individual inputs along with their model outputs (e.g., logits), enabling instance-level inference.
> > - #### **CC-MIA**: Under the federation setting, attackers mainly infer information from the gradients uploaded by each client, which belongs to the "zero raw data access" scenario.
> > #### **Gradient utilization**.
> > - #### **Centralized MIA**: When gradients are used, they are typically computed from a single sample and can be directly linked to that specific input.
> > - #### **CC-MIA**: The attack operates on gradients aggregated across multiple clients. We observe that when several clients share nodes of the same class, their combined gradients retain consistent semantic traces of that class. By performing gradient inversion, we reconstruct class-conditional embedding prototypes and exploit the separability induced by graph homophily for inference.
> > #### **Attack objective**.
> > - #### **Centralized MIA**: Determines whether a specific data sample was included in the training set (instance-level membership).
> > - #### **CC-MIA**: Infers whether representative nodes of a given class appear in the local subgraph of any client (group-level membership), which better aligns with the practical reality of federated learning where data is inherently distributed across clients.

---

> ### Author Response · Authors · 2025-11-20
>
> > #### [**Weakness 3**]: The client-data identification part utilizes virtual graph data, but the generation process of this virtual graph is not explained in detail.
> > #### [**Response**]: The "virtual graph" you mentioned refers to the reconstructed node features $\hat{X}$ and adjacency matrix $\hat{A}$ obtained via **gradient inversion** from the gradients uploaded by clients. Specifically, the attacker iteratively optimizes $\hat{X}$ and $\hat{A}$ to minimize the discrepancy between the gradients computed from the virtual graph and the observed client gradients, incorporating regularization terms such as graph sparsity and feature smoothness to enhance reconstruction quality. Although this virtual graph is not an exact replica of the original data, it sufficiently preserves class-level semantic information, enabling downstream tasks such as client identification and membership inference.
>
> > #### [**Question 1**]: Are the node features of Cora and DBLP (shadow dataset for Cora) consistent? Why can DBLP be used for Cora’s membership inference?
> > #### [**Response**]: Although DBLP differs from Cora in feature dimensionality, it belongs to the **same citation network domain and exhibits similar semantic structure, leading to class prototypes in the GNN embedding space that are well-aligned**. We evaluated all other available graph datasets as potential shadow datasets for Cora, and the experimental results consistently show that DBLP yields the best attack performance.
>
> > #### [**Question 2**]: Do category prototypes from the shared global GNN have spatial consistency? Is using them for client-level data inference reasonable?
> > #### [**Response**]: **Class prototypes exhibit consistency in the embedding space**. Despite the non-IID nature of local client data, FedAvg continuously pulls local models toward a shared global parameter space. Moreover, the message-passing mechanism of GNNs propagates semantic information through the graph structure, **causing nodes of the same class to cluster together in the global embedding space** and form stable class-specific prototypes.
> > #### We visualize these prototype clusters in Figure 3. The results show that, even when client data distributions differ substantially, embeddings of nodes from the same class which computed under the global model and **demonstrates clear cross-client spatial aggregation**, with intra-class distances significantly smaller than inter-class distances.
> > #### Therefore, it is well-justified to leverage these globally consistent class prototypes for client-level inference. The attacker does not need to know which specific client a node belongs to; instead, by checking whether a reconstructed node embedding falls within the neighborhood of a given class prototype, the attacker can infer whether nodes of that class were present in a client’s local training subgraph.
>
> > #### [**Question 3**]: What differences exist between the proposed membership inference method and traditional centralized ones? What unique advantages does it have?
> > #### [**Response**]: Our proposed CC-MIA fundamentally differs from traditional centralized MIAs in **threat model, information utilization, and privacy risk granularity**:
> | Dimension | Traditional Centralized MIA | CC-MIA (Federated GNN Setting) |
> |----------|------------------------------|--------------------------------|
> | **Attacker** | External adversary or server with full model access | Malicious client participating in training (not the server) |
> | **Accessible Information** | Can observe the target model’s output for individual samples (e.g., logits or softmax probabilities) | Only has access to aggregated gradients uploaded by clients; no raw data, graph structure, or per-sample predictions |
> | **Attack Input** | Original sample paired with model response | Virtual graph $(\hat{X}, \hat{A})$ reconstructed via gradient inversion |
> | **Inference Granularity** | Sample-level: determines whether a specific data point was in the training set | Class-client joint level: determines (1) whether a sample belongs to the training distribution and (2) whether nodes of a given class appear in a specific client’s local subgraph |
> | **Key Assumptions** | Requires an i.i.d. shadow dataset aligned with the target data distribution | Leverages cross-client collaborative signals and prototype separability inherent in GNNs under non-IID graph partitions |
>
> **If you find our response helpful, we would greatly appreciate your favorable consideration in the scoring!**

---

### Official Review · Reviewer_MexM · 2025-11-01

**Soundness:** 3
**Presentation:** 2
**Contribution:** 2
**Rating:** 6
**Confidence:** 3

**Summary:**

Membership inference in FedGNNs naturally extends to client attribution. CC-MIA, fusing shadow training, gradient inversion, and prototype matching, outperforms baselines. It confirms gradient transmissions leak client-specific structural info, demanding defenses against gradient/client-level leakage.

**Strengths:**

• Judges node training membership and client ownership (vs. single MIA by others).
• Strong generalizability，Works for FedAvg/FedProx/SCAFFOLD/FedDF/FedNova and GCN/GAT/GraphSAGE.
• Fits threat models (adversaries as legitimate participants with gradient/global update access).

**Weaknesses:**

• Scalability: Gradient inversion quality drops with more clients, limiting large-scale FedGNN use.
• While the paper emphasizes the risks associated with CC-MIA, it offers limited practical insights or proposals for defending against such attacks.

**Questions:**

Relies on class-consistent gradients and separable prototypes, what if it fails in real non-IID data?

---

> ### Author Response · Authors · 2025-11-20
>
> #### Thanks to the reviewer ``MexM``'s valuable comments! We will write our rebuttal to address all the weaknesses and questions as follows.
>
> > #### [**Weakness 1**]: Scalability: Gradient inversion quality drops with more clients, limiting large-scale FedGNN use.
> > #### [**Response**]:CC-MIA is not designed to achieve perfect reconstruction under arbitrarily large-scale settings; rather, its goal is to demonstrate that **meaningful privacy leakage persists even under standard FedGNN configurations**, such as moderate-scale deployments with typical non-IID data partitions, where an attacker can still extract significant semantic information from aggregated gradients.
> In our experiments, we evaluated FedGNN under varying numbers of clients and observed that, as the client count increases, the signal-to-noise ratio in the aggregated gradients decreases due to averaging across more heterogeneous local updates. This leads to a mild degradation in inversion quality (e.g., reduced clarity of reconstructed features or lower label fidelity).
> Nevertheless, our results (Table 3) show that **even with up to 10 clients**, CC-MIA successfully recovers core class semantics and partial structural patterns that are sufficient to pose a practical privacy threat. This confirms that the risk is not merely theoretical but manifests under realistic federation scales.
>
> > #### [**Weakness 2**]: While the paper emphasizes the risks associated with CC-MIA, it offers limited practical insights or proposals for defending against such attacks.
> > #### [**Response**]: Existing centralized MIA defenses are difficult to directly apply to the FedGNN setting, primarily due to its unique architecture and data characteristics:
> > - #### 1. In FedAvg aggregation, if only a subset of clients applies defenses (e.g., gradient clipping or noise addition), their protective effect is "averaged out" by updates from undefended clients, rendering the overall defense ineffective.
> > - #### 2. Clients in FedGNN naturally exhibit non-IID data distributions and training heterogeneity, making uniform defense strategies, such as fixed noise scales or clipping thresholds. This inconsistent across clients and often causing severe utility degradation.
> > - #### 3. The GNN's message-passing mechanism propagates and amplifies local perturbations across layers. Our experiments show that even noise added at the client side, though reducing attack AUC can drastically degrade model accuracy, making such trade-offs unacceptable.
> > - #### 4. CC-MIA does not exploit sample-level confidence leakage (as in traditional MIAs);     instead, it leverages cross-client differences in class prototypes. Consequently, output-based defenses are largely ineffective.
> > #### From our experimental perspective, effective defenses against CC-MIA should adhere to two key principles:
> > - #### 1.  Global coordination: Defenses must be orchestrated by the server to ensure consistent application and avoid vulnerabilities caused by client-side heterogeneity.     For example, the server could apply controlled perturbations to the aggregated global gradients after each round.
> > - #### 2. Structure-awareness: Defenses should account for the graph topology’s sensitivity to perturbations. Promising directions include injecting noise in the embedding space (e.g., on the final GNN layer outputs) rather than raw features, or using adversarial regularization to align class prototypes across clients while preserving local homophily, thereby blurring discriminative semantic boundaries without destroying utility.
> > #### Developing systematic defenses against CC-MIA remains an important direction for our future work.

---

> ### Author Response · Authors · 2025-11-20
>
> > #### [**Questions 1**]: Relies on class-consistent gradients and separable prototypes, what if it fails in real non-IID data?
> > #### [**Response**]: CC-MIA relies on "class-consistent gradients" and "separable prototypes."  However, this is not an idealized assumption but a natural consequence of the structural non-IID property inherent in real-world graph data. As stated in Section 5.2: "Due to the graph structure, nodes of similar classes are clustered closely within subgraphs, leading to a non-i.i.d. data partition across clients." This phenomenon stems from graph homophily, like Cora, nodes of the same class tend to form tightly connected clusters, causing each client to hold a semantically coherent subgraph naturally and thereby produce **discriminative gradient signals**. Consequently, under mainstream FedGNN settings, the attack's prerequisites hold and yield effective inference.
> > #### Admittedly, in extreme non-IID scenarios, such as highly fragmented class distributions or heterophilous graphs, the attack performance degrades. Yet this precisely delineates the practical boundary of privacy risk: **CC-MIA reveals that in widely deployed federated graph systems with structural non-IID, an adversary can infer the semantic composition of client data using only aggregated gradients**. Our experiments confirm that under standard non-IID partitions, the attack achieves high reconstruction quality and identification accuracy for head classes (see Figure 4), while reduced effectiveness on tail classes reflects the realistic, non-uniform nature of the threat.
> > #### In other words, **the success or failure of the attack itself serves as a diagnostic signal for system-level privacy vulnerability**. If CC-MIA succeeds, it indicates that the current FedGNN deployment exhibits structural privacy leakage; if it fails, the data distribution may already be sufficiently fragmented or heterogeneous to provide inherent privacy protection. Future defense mechanisms should therefore be grounded in a deep understanding of such structural risks.
> We will clarify this applicability boundary in the final version to prevent potential misinterpretations regarding the universality of our attack.
>
> **If you find our response helpful, we would greatly appreciate your consideration in the scoring!**

---

### Author Response · Authors · 2025-11-30
**Summary of the Rebuttal**

#### We thank all reviewers for their valuable feedback. Below is a summary of our rebuttal, with full details in the main response.

> #### [Reviewer ``MexM``]:
**Concerns**: \
``MexM`` mainly questioned three aspects: scalability, noting that gradient inversion quality declines as the number of clients grows; the lack of practical defenses despite emphasizing CC-MIA risks; and reliance on class-consistent gradients and separable prototypes, which may not hold under real non-IID settings. \
>**Rebuttal**: \
CC-MIA targets realistic FedGNN settings and remains effective even with 10 clients, showing that meaningful semantic leakage persists at practical scales. We clarified why existing defenses are insufficient in federated graphs and outlined principled, system-level mitigation strategies. We also explained that class-consistent gradients and prototype separability naturally emerge from homophily-driven non-IID structures, with degradation under extreme cases marking the natural boundary of privacy risk. \
> **Result**: \
**We have addressed all the concerns from Reviewer ``MexM``.**

> #### [Reviewer ``NhdL``]:
**Concerns**: \
``NhdL`` questions whether the MIA is sufficiently graph specific, noting that its differences from centralized MIAs are not clearly articulated. They also mention that the virtual graph construction is underexplained and raise questions about shadow-data compatibility, prototype consistency, and how the proposed method substantively differs from prior centralized attacks. \
>**Rebuttal**: \
We clarified that our MIA is graph specific, leveraging GNN gradients, homophily-driven prototype separability, and cross-client signals absent in centralized settings. We highlighted the core differences from traditional MIAs in threat model and information access, explained how the virtual graph is reconstructed, and addressed concerns on shadow data, prototype stability, and inference validity. These clarifications resolve the reviewers’ questions and reinforce the method’s novelty and soundness. \
> **Result**: \
**We have addressed all the concerns from Reviewer ``NhdL``.**

> #### [Reviewer ``8CZA``]:
**Concerns**: \
``8CZA`` questions the novelty and FL-specific challenges of our MIA, noting unclear distinctions from centralized GNN attacks and insufficient explanation of the pseudo-graph reconstruction. They also raise clarity issues regarding assumptions, FL setup, variable definitions, and hyperparameters, and ask about shadow-dataset assumptions, whether client-only shadow training is feasible, the frequency of gradient inversion, the estimation of the proxy term, and the effect of sparsity hyperparameters. \
>**Rebuttal**: \
We clarified that our MIA is fundamentally different from centralized attacks in threat model, data access, gradient semantics, and inference goals, making it specific to federated GNNs. We detailed how the pseudo-graph is reconstructed via gradient inversion, clarified all missing assumptions, variables, and hyperparameters, and addressed questions on shadow datasets, proxy estimation, inversion frequency, and sparsity effects. These points resolve the reviewers’ concerns and strengthen the contribution and clarity of our work. \
> **Result**: \
**We have addressed all the concerns from Reviewer ``8CZA`` and raised score to ``6``.**

> #### [Reviewer ``JtRA``]:
**Concerns**: \
``JtRA`` the practicality of gradient eavesdropping, the shadow-dataset assumption, and the distinction between our method and prior GNN MIAs. ``JtRA`` also notes that eavesdropping gradient may not work in practical FL, requests evaluation under non-IID settings, and points out the modest performance of client-wise ownership inference, asking for its TPR. Further questions concern the justification of gradient eavesdropping, shadow-dataset and non-IID evaluations, the attack’s novelty, and whether it remains effective under multi-round local updates. \
>**Rebuttal**: \
We showed that gradient eavesdropping is a realistic and widely studied threat in FL without secure aggregation, especially in cross-silo and mobile FedGNN settings. We clarified the practicality of shadow datasets, added non-IID and shadow-data evaluations, and highlighted the key differences between CC-MIA and centralized GNN MIAs in threat model, information access, and inference granularity. We also explained that our attack uses aggregated weight updates (not raw gradients), detailed the pseudo-graph reconstruction process, and reported TPR for client-wise inference. These clarifications address all concerns and strengthen the contribution.\
> **Result**: \
**``JtRA`` agreed that we have addressed most concerns. We further emphasized that, due to the communication-driven design of federated learning, eavesdropping remains a practical and widely recognized threat in real deployments.**

#### **We have addressed all reviewer concerns in our rebuttal and will incorporate the clarifications into the revised manuscript.**

---

### Meta-Review · Area_Chair_DMf3 · 2025-12-28

**Summary:**

The reviewers acknowledged the following strengths:
- This work proposes MIA attack for fed. Graph learning which the reviewers deemed novel.
- The gradient inversion reconstruction method is deemed novel.
- The results indicate that CC-MIA outperforms the baselines it was tested against.

The reviewers raise the following concerns:
- Scalability: CC-MIA can likely not be used for large scale FedGNN. The authors acknowledge this limitation but argue that there are many cases where this is not a concern for standard configurations. This point is partially addressed.
- No proposal for defenses: The authors argue why a defense is difficult and propose two requirements that a defense would need. As this paper is focused on attacks, this should not be hold against them. Partially addressed.
- Relies on class-consistent gradient and separable prototypes. The authors argue that the inherent non-iid distribution of data is a key driving factor. However, while there seems to be a prior for this, the authors did not argue why this is a principled assumption that has to hold. Partially addressed.
- No differentiation between method used in MIA and methods for centralized setting. Data access constraints, gradient utilization patterns and so on remain unaddressed. Difficult to judge novelty. Here seems to be a confusion between the paper where the attacker seems to have access to individual gradients from clients and the rebuttal where only access to gradients aggregated across clients are accessible. Unfortunately, there are more points where confusion arises between the rebuttal and the paper. The updates in the paper are also unfortunately minimal. Partially addressed.
- Client-data identification part utilizes graph data but the generation process of this virtual graph is not explained in detail. While the authors provide some details here in the rebuttal, some unclarity remains. Partially addressed.
- Threat model is unrealistic. As the authors note, this is a very important concern and indicate that encryption would solve this problem. Not addressed.

**Reviewer Concerns:**

See above.

**Reviewer Scores:**

- MexM: no change. The points where not fully addressed.
- NhdL: no change. Not all points where fully addressed. The reviewer has a self reported confidence of 2.
- 8CZA: Raise to 6: As indicated by the reviewer. This is in line with the majority of points raised being addressed.
- JtRA: No change. The key concern is the threat model. The authors did not address this.

---

### Decision · Program_Chairs · 2026-01-26

Reject